# Single cell RNA-seq identifies the origins of heterogeneity in efficient cell transdifferentiation and reprogramming

**Mirko Francesconi[1†‡], Bruno Di Stefano[1,2†], Clara Berenguer[1], Luisa de Andrés-Aguayo[1], Marcos Plana-Carmona[1], Maria Mendez-Lago[3§], Amy Guillaumet-Adkins[3#], Gustavo Rodriguez-Esteban[3], Marta Gut[3], Ivo G Gut[3], Holger Heyn[3], Ben Lehner[1,4*], Thomas Graf[1,5*]**

[1]Centre for Genomic Regulation (CRG), The Barcelona Institute of Science and Technology (BIST), Barcelona, Spain; [2]Department of Stem Cell and Regenerative Biology, Harvard University, Cambridge, United States; [3]CNAG-CRG, Centre for Genomic Regulation (CRG), The Barcelona Institute of Science and Technology (BIST), Barcelona, Spain; [4]Institució Catalana de Recerca i Estudis Avançats (ICREA), Barcelona, Spain; [5]Universitat Pompeu Fabra (UPF), Barcelona, Spain

**\*For correspondence:**
lehner.ben@gmail.com (BL);
Thomas.Graf@crg.eu (TG)

[†]These authors contributed equally to this work

**Present address:** [‡]Laboratoire de Biologie et Modélisation de la Cellule (LBMC), Universite Lyon, Ecole Normale Superieure de Lyon, CNRS UMR 5239, INSERM U 1210, Universite Claude Bernard Lyon 1, Lyon, France; [§]Institute of Molecular Biology, Mainz, Germany; [#]Department of Pediatrics, Dana Farber Cancer Institute, Harvard Medical School, Boston, United States

**Abstract** Forced transcription factor expression can transdifferentiate somatic cells into other specialised cell types or reprogram them into induced pluripotent stem cells (iPSCs) with variable efficiency. To better understand the heterogeneity of these processes, we used single-cell RNA sequencing to follow the transdifferentation of murine pre-B cells into macrophages as well as their reprogramming into iPSCs. Even in these highly efficient systems, there was substantial variation in the speed and path of fate conversion. We predicted and validated that these differences are inversely coupled and arise in the starting cell population, with Myc[high] large pre-BII cells transdifferentiating slowly but reprogramming efficiently and Myc[low] small pre-BII cells transdifferentiating rapidly but failing to reprogram. Strikingly, differences in Myc activity predict the efficiency of reprogramming across a wide range of somatic cell types. These results illustrate how single cell expression and computational analyses can identify the origins of heterogeneity in cell fate conversion processes.
DOI: https://doi.org/10.7554/eLife.41627.001

## Introduction

Elucidating the transcriptional programs that determine cell identity during development and regeneration is one of the major goals of current stem cell research. In the past decade, several groups have demonstrated cell plasticity, meaning that a variety of somatic cells can be converted into either pluripotent cells or into other specialised cells by overexpression of specific transcription factors (TFs) (*Graf and Enver, 2009*; *Jopling et al., 2011*). For example, the Yamanaka factors *Pou5f1*, *Sox2*, *Klf4* and *Myc* (OSKM) can reprogram somatic cells into induced pluripotent stem cells (iPSCs) (*Takahashi and Yamanaka, 2006*), while lineage-instructive TFs can prompt the transdifferentiation of mouse and human cells into other specialised cell types such as muscle, neural or hematopoietic cells (*Vierbuchen et al., 2010*; *Xie et al., 2004*; *Davis et al., 1987*; *Graf, 2011*). In all cases one gene expression program is erased and a new one established. Typically only a small fraction of cells successfully acquire a new fate after TF-overexpression (*Hochedlinger and Plath, 2009*). For instance, the efficiency of conversion into iPSCs in response to OSKM of diverse primary adult cells such as fibroblasts, keratinocytes, liver cells, neural precursor cells, pancreatic β cells and granulocyte/macrophage progenitors (GMPs) varies widely, ranging between 0.01% for T-lymphocytes and

25% for GMPs (*Eminli et al., 2009*; *Kim et al., 2008*; *Stadtfeld et al., 2008*; *Aoi et al., 2008*; *Aasen et al., 2008*) for unclear reasons. Identifying the transcriptional signature that render a somatic cell type more amenable to transdifferentiation or reprogramming would teach us about the general mechanisms that control cell fate.

Mechanistic studies of transdifferentiation and reprogramming have established that these are complex processes, where multiple players synergistically establish new transcriptional networks, disrupt old ones and remove epigenetic barriers (*Buganim et al., 2013*). Among the factors that have been shown to affect the efficiency and kinetics of reprogramming are proteins involved in cell cycle progression, chromatin remodelling, and posttranscriptional regulation (*Di Stefano et al., 2016*; *Krizhanovsky and Lowe, 2009*; *Chen et al., 2013*; *Doege et al., 2012*; *Cheloufi et al., 2015*; *Brumbaugh et al., 2018*; *Li et al., 2017*; *van Oevelen et al., 2015*; *Wapinski et al., 2013*). Despite these insights, many details about cell fate conversion processes remain unclear. Do cells convert fates as homogeneous populations or through a diversity of paths? Do all cells convert with the same speed? And what are the determinants of variation in the speed and path of cell fate conversion? More fundamentally, if an individual cell is more susceptible for conversion into one fate, is it also more susceptible to conversion into alternative fates? Major obstacles to tackling these questions are the use of bulk samples for analysis, which obscures transcriptional variability in both the starting cell population and during fate conversion, as well as the typically small proportion of responding cells.

To overcome these bottlenecks we employed high-throughput single cell RNA-sequencing (MARS-Seq (*Jaitin et al., 2014*)) to analyse two highly efficient cell conversion protocols applied to the same starting cell population: i) the transdifferentiation of pre-B cells into macrophages induced by the TF C/EBPa (*Xie et al., 2004*) and, ii) the reprogramming of pre-B cells into iPSCs, based on the transient expression of C/EBPα followed by the induction of OSKM (*Di Stefano et al., 2014*). This revealed that both processes, despite their very high efficiency, show heterogeneity for the speed and path of cell fate conversion: cells do not all convert at the same rate and along the same path to the two terminal fates. We computationally predicted and experimentally validated that this heterogeneity arises in the starting cell population. Cells with low Myc activity transdifferentiate into macrophages efficiently and directly but fail to reprogram. In contrast, cells with high Myc activity reprogram at a very high efficiency but have a lower propensity to transdifferentiate and do so by a more indirect path. Strikingly, Myc levels correlate with the reprogramming efficiency of diverse hematopoietic and non-hematopoietic cell types. These results illustrate how single cell analysis can characterise heterogeneity in cell fate conversion processes and identify its underlying causes.

## Results

### Single cell analysis of highly efficient transdifferentiation and reprogramming from the same cell population

We isolated CD19+ pre-B cells from the bone marrow of reprogrammable mice carrying a drug-inducible reverse tetracycline trans-activator (M2rtTA; hereafter abbreviated as rtTA) in the *Rosa26* locus, a polycistronic expression cassette in the collagen type I (*Col1a1*) locus, which contains four mouse derived cell reprogramming genes (*Pou5f1*, *Sox2*, *Klf4* and *Myc*, OSKM) separated by three sequences encoding 2A self-cleaving peptides, and the POU5F1-*GFP* transgene (*Di Stefano et al., 2014*; *Carey et al., 2010*). Pre-B cells were then infected with a C/EBPaER-hCD4 retrovirus, sorted for hCD4 expression and induced to either transdifferentiate into macrophages or reprogram into iPSCs. To induce the macrophage fate, we treated the cells with beta-estradiol (E2), which activates C/EBPα. To induce the iPSC fate, we first incubated them with E2 for 18 hr to transiently activate C/EBPa, generating a 'poised state', washed out the compound and then added doxycycline to induce OSKM (*Di Stefano et al., 2016*; *Di Stefano et al., 2014*). For transdifferentiation, we collected cells before (0 hr) and after 6 hr, 18 hr, 42 hr, 66 hr and 114 hr of C/EBPα induction; for reprogramming, samples were prepared at days 2, 4, 6 and 8 after OSKM induction of 18 h C/EBPa-pulsed cells (*Figure 1a*), to be consistent with our previous bulk studies (*Di Stefano et al., 2016*; *Di Stefano et al., 2014*). We collected two pools of 192 cells at each time point and sequenced their RNA using MARS-Seq (*Jaitin et al., 2014*). After quality control and filtering, we obtained expression profiles for 17,183 genes in 3,152 cells. After performing dimensionality reduction and correction for global

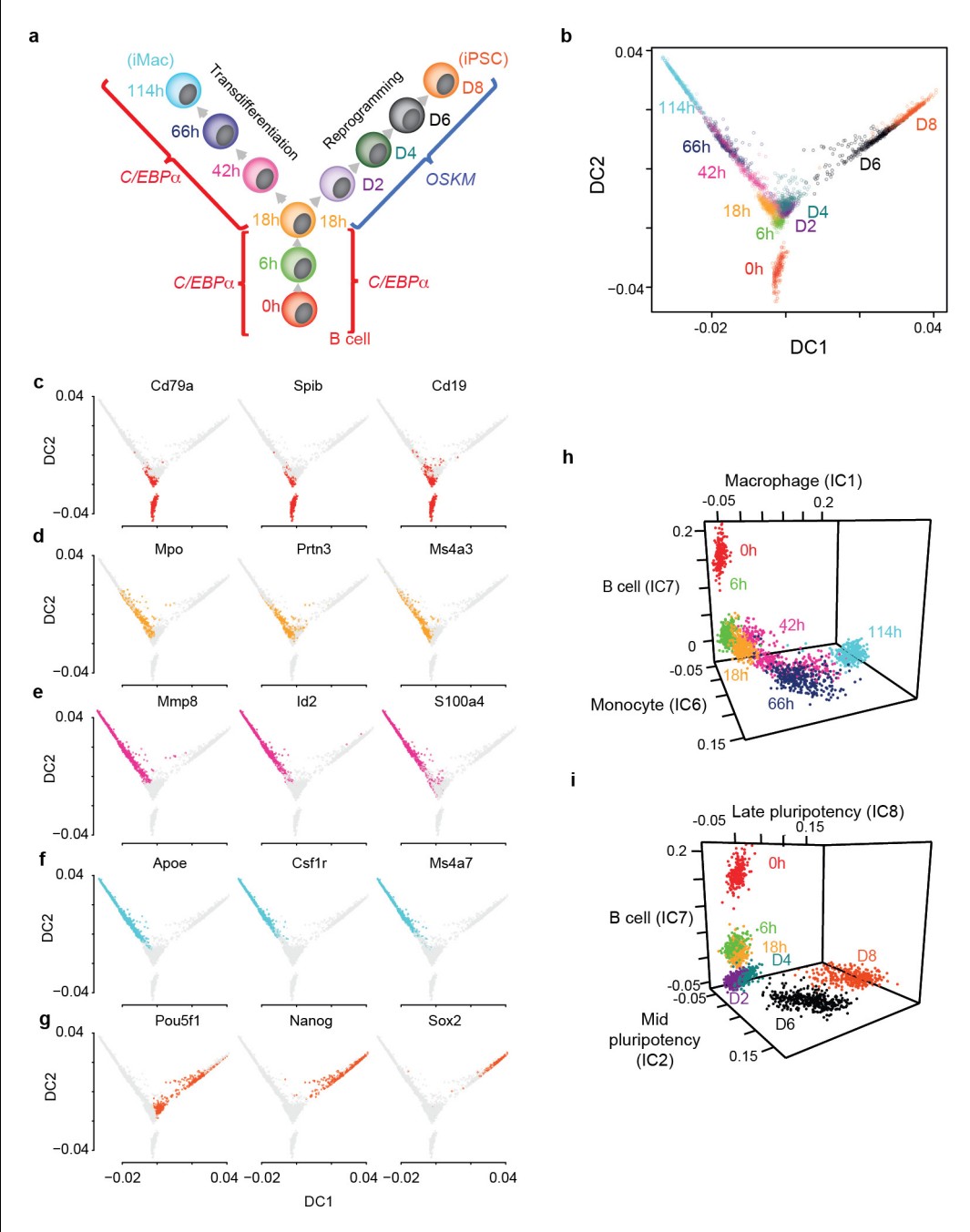

**Figure 1.** Single cell gene expression analysis of B cell to macrophage transdifferentiation and B cell to iPSC reprogramming. (**a**) Overview of the experimental design, showing time points analysed. (**b**) Single cell projections onto the first two diffusion components (DC1 and DC2). c-f, as in b, with top 50% of cells expressing selected markers for B cells in red (**c**) GMP/granulocytes in orange (**d**) monocytes in purple (**e**) macrophages in light blue (**f**) and pluripotent cells in orange-red (g). (**h-i**) Projection of transdifferentiating cells onto B cell-, macrophage-, and monocyte-specific independent components (**h**) and reprogramming cells onto, B cell-, mid- and late-pluripotency specific independent components as defined in *Figure 1—figure supplement 2a* (**i**).

DOI: https://doi.org/10.7554/eLife.41627.002

The following figure supplements are available for figure 1:

**Figure supplement 1.** Data pre-processing, batch correction and independent component analysis.

DOI: https://doi.org/10.7554/eLife.41627.003

*Figure 1 continued on next page*

*Figure 1 continued*

**Figure supplement 2.** Characterisation of independent components, gene expression reconstruction and diffusion maps.
DOI: https://doi.org/10.7554/eLife.41627.004
**Figure supplement 3.** Single cell analysis of reprogramming and transdifferentiation.
DOI: https://doi.org/10.7554/eLife.41627.005
**Figure supplement 4.** Gene expression distribution of markers during reprogramming and transdifferentiation.
DOI: https://doi.org/10.7554/eLife.41627.006

batch effects (*Figure 1—figure supplement 1a–c*) we used independent component analysis (ICA) to extract specific gene expression signatures (*Supplementary files 1–2* and *Figure 1—figure supplement 1d*). We then compared independent components from our data with components extracted from a comprehensive atlas of 272 murine cell types (*Hutchins et al., 2017*) (*Figure 1—figure supplement 2a–c*). The components were further characterised by Fisher's test-based gene set enrichment analysis (GSE, *Supplementary files 3–4*). Finally, we reconstructed batch corrected gene expression data using selected components (*Figure 1—figure supplement 2d–e*).

## Cell conversion trajectories suggest that transcription factor induced transdifferentiation and reprogramming are deterministic

Visualising the data using diffusion maps (*Haghverdi et al., 2015*) revealed branching between transdifferentiation and reprogramming at the 18 hr time-point, with cohorts of cells moving along two distinct trajectories and reaching homogeneous final cell populations consisting of either induced macrophage (iMac) or iPSC-like cells, respectively (*Figure 1b*, *Figure 1—figure supplement 2f*, *Figure 1—figure supplement 3a*). We observed no branching into alternative routes, in contrast to what has been described for the transdifferentiation of fibroblasts into neurons (*Treutlein et al., 2016*), muscle cells (*Cacchiarelli, 2017*) or iPSCs (*Guo, 2017*; *Schiebinger, 2017*). Our findings therefore support the notion that both transcription factor-induced transdifferentiation and reprogramming represent deterministic processes. However, we observed that D2-D4 cells transit through a state that partially resembles neuronal cell types (*Figure 1—figure supplement 3a*) although the significance of this is unclear as D2 and D4 cells are overall quite dissimilar to any cell type within the mouse cell reference atlas (*Hutchins et al., 2017*) (*Figure 1—figure supplement 3b*). Of note, D6 cells are more similar to inner cell mass cells (ICM) at the blastocyst stage than D8 cells, which are in turn more similar to ESCs (*Figure 1—figure supplement 3a*). This observation is reminiscent of recent findings showing that cells at intermediate stages of mouse embryo fibroblast (MEF) to iPSC reprogramming exhibit an increased ability to generate diverse somatic tissues upon injection into tetraploid blastocysts (*Amlani et al., 2018*).

## C/EBPa expression silences the B cell program and induces a progenitor state followed by a monocyte/macrophage program

Already 6 h hours after C/EBPa expression cells strongly downregulated B cell-specific transcripts, such as *Cd19* that encodes a B lineage transmembrane protein; *Cd79a*, *Cd79b*, *Vpreb1*, *Vpreb2*, *Vpreb3* and *Blnk* that are involved in signalling of the B cell receptor complex; and *Blk* that encodes a B lymphocyte specific kinase. Subsequently, after 18 hr they started to transiently express the granulocyte/GMP restricted genes myeloperoxidase (*Mpo*) and the serine neutrophil protease 3 (*Prtn3*). Finally, after sustained C/EBPa expression additional myeloid markers become expressed, including the macrophage specific colony stimulating receptor gene (*Csf1r*), lysozyme (*Lyz1 and 2*), granulocyte collagenase 8 (*Mmp8*), macrophage scavenger receptor (*Msr1*), myeloid restricted serine protease C (*Ctsc*) and the myeloid cytokine dependent chemokine 6 (*Ccl6*) (*Figure 1c–f*, *Figure 1—figure supplement 3c–j*, *Supplementary files 6* and *7*).

## OSKM expression in C/EBPa-pulsed cells further accelerates B cell silencing and leads to the sequential upregulation of the pluripotency program

After OSKM induction of 18 h C/EBPa-pulsed cells, endogenous *Pou5f1* (*Oct4*) is activated at day 2, followed by expression of *Nanog* and *Sox2* at days 6 and 8, respectively (*Figure 1g*, *Figure 1—*

*figure supplement 3g–i*). This is consistent with the sequential expression of the three key pluripotency factor genes revealed by RNA sequencing of bulk populations during reprogramming in our system (*Di Stefano et al., 2016*; *Stadhouders et al., 2018*). OSKM induction further downregulates B cell genes and inflammatory genes and upregulates biosynthetic pathway and energy metabolism genes at D2 (*Figure 1—figure supplement 3k*, *Supplementary files 6–7*). This is followed by activation of proliferation and cell cycle genes at D6 and histone deacetylase and methylase genes at D8 (*Figure 1—figure supplement 3l*, *Supplementary files 6–7*. In short, OSKM expression in C/EBPa pulsed cells further induces B cell silencing and activates pluripotency genes in a sequential manner.

## Heterogeneity at intermediate time points suggests asynchrony in transdifferentiation timing

Visualising single cells in diffusion maps (*Figure 1b*) or in the expression space spanned by B cell, monocyte and macrophage programs (*Figure 1h*) shows that at intermediate time points some cells are highly similar to cells at earlier time points while others are similar to cells at later time points. For example, at 42 hr after C/EBPa induction there are three clusters of cells that are spread along the transdifferentiation trajectory (*Figure 1h*, magenta). Consistently, the expression of key marker genes at 42 hr of transdifferentiation is highly variable, with some cells expressing levels comparable to cells at earlier time points and others expressing levels comparable to cells at later time points (*Figure 1—figure supplement 4a*). These observations suggest heterogeneity in the speed of transdifferentiation conversion (i.e. asynchrony) among single cells despite the fact that transdifferentiation results in a quite homogeneous final cell population.

## Rapid transdifferentiation into macrophages is associated with low Myc component

To identify potential causes of this apparent asynchronous behaviour, we define each cell progression towards a macrophage state as the genome-wide similarity of its transcriptome to the bone-marrow-derived-macrophage (BMDM) transcriptome from the reference atlas (*Hutchins et al., 2017*) (see Methods). We can observe also using this metric that at 42 hr some cells already resemble macrophages while others are still quite dissimilar, which is consistent with asynchrony in transdifferentiation (*Figure 2a*). We then determined which gene expression signature extracted from our single cell expression data correlates best with the progression towards the macrophage state (*Figure 2a*) at each time-point (excluding the cell type-specific signatures directly involved in transdifferentiation, that are the B cell, monocyte, granulocyte, and macrophage programs). The most correlated component is highly enriched in Myc target genes (component five in *Figure 1—figure supplement 2*, henceforth called 'Myc component', see Fisher's test-based enrichment analysis of Molecular Signature database hallmark gene set collection (*Liberzon et al., 2015*) in *Supplementary file 4*) and negatively correlates with the progression of cells at intermediate time points of transdifferentiation (*Figure 2b*, *Figure 2—figure supplement 1*). The Myc component varies extensively across cells within each time point but overall changes little during transdifferentiation (*Figure 2c*). These data therefore suggest that cells with lower Myc component transdifferentiate more rapidly into macrophages.

## Cells with high Myc component transdifferentiate via a pronounced GMP-like cell state

We next tested how the Myc component relates to the loss of the B cell state during transdifferentiation. For each cell's transcriptome, we therefore computed its similarity to the pre-B cell state and compared this with its similarity to the macrophage state. This shows that low Myc component is more strongly associated with a rapid gain of the macrophage state than with a rapid loss of the B cell state (the cells go from high Myc to low Myc component mainly from left to right along the macrophage axis rather than from top to bottom along the pre-B cell axis; *Figure 2d*). Similarly, we explored the acquisition of a transient GMP-like state during transdifferentiation. This shows that high Myc component cells (green/blue) resemble GMPs up to 42 hr after C/EBPa induction whereas low Myc component cells (yellow/orange) only show moderate similarity to GMPs at 18 hr after induction, suggesting that higher Myc component is associated with a larger and more persistent induction of a GMP-like state (*Figure 2e*). However, this is not the case for the induction of a

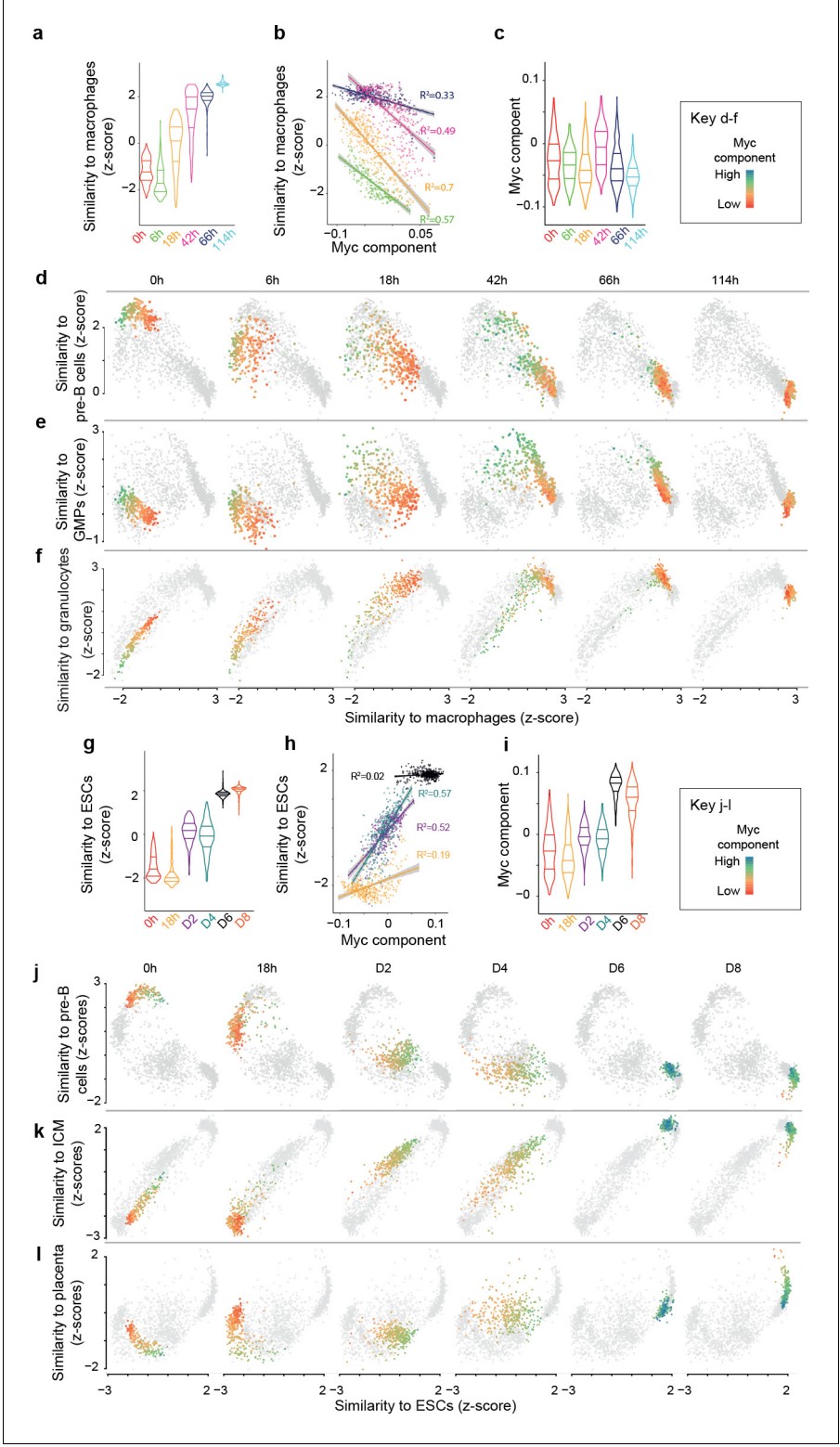

**Figure 2.** Myc activity correlates with differences in single cell transdifferentiation and reprogramming trajectories. (a) Distribution of gene expression similarity between single cells and reference bone marrow derived macrophages (*Hutchins et al., 2017*) (acquisition of macrophage state) during transdifferentiation. (b) Correlation between the Myc component and acquisition of macrophage state from a; start and end time points were omitted to improve clarity (they are presented in *Figure 2—figure supplement 1a*). (c) Myc component at the various transdifferentiation time points. d-f, Single cell trajectories of the B cell state (d), the GMP state (e) and the

*Figure 2 continued on next page*

*Figure 2 continued*

granulocyte state (f) related to the acquisition of the macrophage state during transdifferentiation. The cells at the respective time points are coloured according to Myc component levels. (g) Distribution of expression similarity between single cells and reference embryonic stem cells (ESCs) during reprogramming. (h) Correlation between Myc component and acquisition of pluripotency from g. (i) Myc component at the various reprogramming time points. (j-l) Single cell trajectories of the B cell state (j), GMP state (k) and inner cell mass state (l) related to the acquisition of the pluripotent state (ESCs) (see also *Figure 3—figure supplement 1*).

DOI: https://doi.org/10.7554/eLife.41627.007

The following figure supplements are available for figure 2:

**Figure supplement 1.** Predicting the speed of transdifferentiation.
DOI: https://doi.org/10.7554/eLife.41627.008
**Figure supplement 2.** Predicting the speed of reprogramming.
DOI: https://doi.org/10.7554/eLife.41627.009
**Figure supplement 3.** High Myc component correlates with faster route towards reprogramming also when factoring out Myc component and cell cycle components before the computation of the similarity score.
DOI: https://doi.org/10.7554/eLife.41627.010

transient granulocyte-like state (*Figure 2f*). Taken together, these analyses suggest that high Myc component cells acquire the macrophage fate more slowly than low Myc component cells, passing through a more pronounced induction of a GMP-like state.

## Efficient reprogramming correlates with high Myc component

Next, as we did for the transdifferentiation, we define each cell progression towards pluripotency as the genome-wide similarity of its transcriptome to the ESC transcriptome from the reference atlas (*Hutchins et al., 2017*) (see Methods). At intermediate time points (D2 and D4) the similarity to ESC is quite variable, with cells that already resemble ESCs and others that are still quite dissimilar (*Figure 2g*), suggesting an asynchronous behaviour during reprogramming as well. We then searched for expression signatures that best correlate with the progression of individual cells toward pluripotency within each time-point during reprogramming. The Myc component again correlates best with progression of cell fate conversion, especially at early stages. However, in contrast to what was observed during transdifferentiation, high Myc component positively correlates with a more advanced state of reprogramming (*Figure 2h*, *Figure 2—figure supplement 2*). Moreover - and also different to what was observed during transdifferentiation - the Myc component increases during reprogramming (*Figure 2i*).

We next explored how the Myc component relates to the loss of B cell program during reprogramming (*Figure 2j*). As for transdifferentiation cells go from low Myc to high Myc component from left to right along the similarity to ESC (x axis) rather than from top to bottom along the similarity to pre-B cells (y axis), suggesting that high Myc component correlates more with the gain of pluripotency rather than with the loss of the B cell program. As mentioned before, D6 cells are more similar to early embryonic stages than D8 cells. However, exploring how Myc component relates to the similarity to inner cell mass (ICM) cells during reprogramming shows that cells with high Myc component maintain a high similarity to ICM cells at D8. In contrast, cells with low Myc component show low similarity to ICM cells at D8 (*Figure 2k*, *Figure 2—figure supplement 3g*). Interestingly, low Myc component cells also acquire a placental-like signature at D8 (*Figure 2l*, *Figure 2—figure supplement 3h*), suggesting that low Myc component cells may eventually branch out towards this extraembryonic lineage.

Together, our findings reveal a correlation between high Myc component and cell susceptibility to reprogramming towards pluripotency. They also suggest that a subset of low Myc component cells along this trajectory acquires properties of extraembryonic cells.

## Variation in Myc component reflects pre-existing variation in the starting cell population

What is the origin of the Myc component heterogeneity? Is it due to a differential response of lineage instructive transcription factors of an essentially homogenous population or to a heterogeneity in the starting population? Examining the uninduced pre-B cells shows a variable Myc component

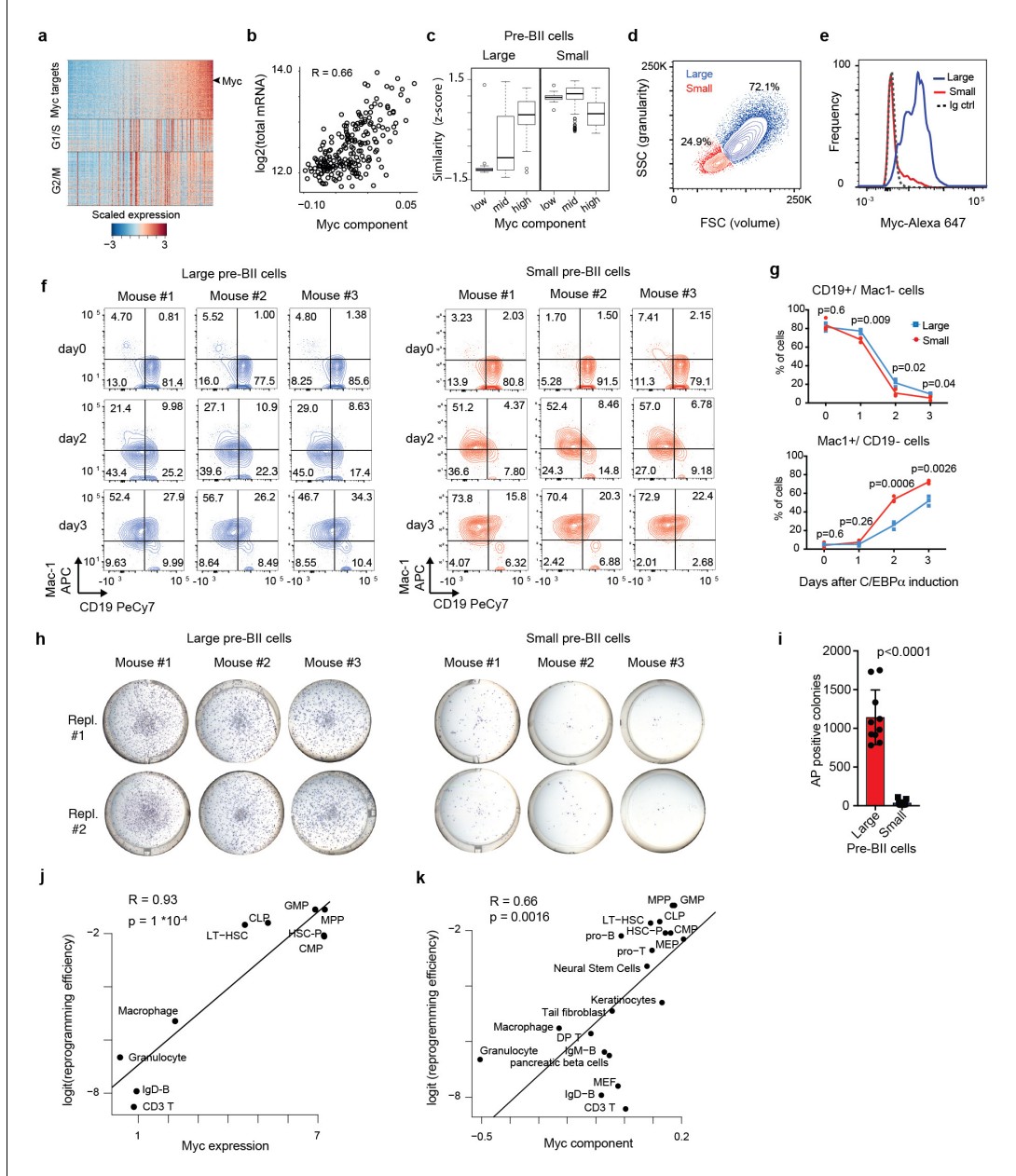

**Figure 3.** Two types of pre-B cells exhibit distinct cell conversion plasticities. (**a**) Heatmap showing the expression of Myc target genes, G1/S and G2/M specific genes in the starting pre-B cells sorted by Myc component. (**b**) Pearson's correlation between total mRNA molecules per cell and Myc component. (**c**) Similarity score of single cells binned by Myc component (bottom 20%, mid and top 20%) with reference large and small pre-BII cells. (**d**) Representative FACS plot of starting pre-B cells showing forward (FSC) and side scatter (SSC). (**e**) Representative FACS analysis of Myc levels detected in the 30% largest and the 30% smallest pre-B cell fractions. (**f**) FACS plots of myeloid marker (Mac-1) and B cell marker (CD19) expression during induced transdifferentiation of sorted large and small pre-BII cells. (**g**) Quantification of the results shown in f (n = 3 biological replicates, error bars indicate mean ± s.d. Statistical significance was determined using multiple t-test with 1% false discovery rate). (**h**) Visualisation of iPSC-like colonies (stained by alkaline phosphatase) 12 days after OSKM induction of sorted large and small pre-BII cells. (**i**) Quantification of the results shown in h (n = 10 biologically independent samples (cell cultures) for large and n = 9 biologically independent samples (cell cultures) for small cells, with error bars indicating mean ±s.d. Statistical significance was determined using a two-tailed unpaired Student's t-test). (**j**) Scatterplot showing the correlation between Myc expression (*Jaitin et al., 2014*) in different starting hematopoietic cell types (x-axis) and their corresponding (logit transformed) reprogramming efficiency (y-axis). GMP: granulocyte monocyte progenitor, CMP: common myeloid progenitor, CLP: common lymphoid progenitor, LT-HSC: long term hematopoietic stem cells, HSC-P: short term hematopoietic stem cells. (**k**) Correlation between Myc component and reprogramming efficiency in various somatic cell types, including the hematopoietic cells shown in j.

DOI: https://doi.org/10.7554/eLife.41627.011

*Figure 3 continued on next page*

*Figure 3 continued*

The following figure supplements are available for figure 3:

**Figure supplement 1.** Experimental data relevant for *Figure 3*.

DOI: https://doi.org/10.7554/eLife.41627.012

**Figure supplement 2.** Gating strategies for FACS analyses.

DOI: https://doi.org/10.7554/eLife.41627.013

which also partially correlates with higher expression of both G1/S and G2/M phase cell cycle genes (*Figure 3a*). Visualising transcriptomes of uninduced pre-B cells using t-SNE indeed revealed substructure associated with the Myc component (*Figure 3—figure supplement 1a*). In addition, the Myc component in our single pre-B cells scales with the total mRNA content of each cell which varies over a three-fold range (*Figure 3b*). This suggests a Myc-associated heterogeneity in cell size in the starting cell population. During B cell development in the bone marrow, large pre-BII cells undergo a proliferation burst and following activation of the pre-B cell receptor differentiate into quiescent small pre-BII cells via Bcl6-induced transcriptional repression of *Myc* (*Hoffmann et al., 2002*). These events constitute an important immunological checkpoint, required for the initiation of light chain immunoglobulin rearrangements (*Nahar et al., 2011*). Thus, we hypothesised that the heterogeneity in the starting pre-B cell population could reflect variability along this B cell developmental transition. To test this hypothesis, we compared our single cell data with bulk expression data of cells at various stages of B cell development (*Hoffmann et al., 2002*; *Painter et al., 2011*). This revealed that cells with higher Myc component are indeed more similar to large and cycling pre-BII cells, while cells with lower Myc component are more similar to small and non-cycling pre-BII cells (*Figure 3c*, *Figure 3—figure supplement 1b*).

Taken together, our analyses suggest a pre-existing heterogeneity in the starting cell population, corresponding to large and small pre-BII cells. Moreover, they suggest that small pre-BII cells should transdifferentiate faster but reprogram more slowly, while large pre-BII cells should transdifferentiate more slowly but reprogram faster.

## Large and small pre-B cells differ reciprocally in their respective transdifferentiation and reprogramming propensities

To test these hypotheses, we analysed our starting pre-B cell population by flow cytometry and found that it can be resolved into two discrete subpopulations, with 72% large and 25% small cells (*Figure 3d*). Intracellular staining of Myc monitored by flow cytometry confirmed that the large cells express abundant levels of the transcription factor while the smaller cells are essentially Myc negative (*Figure 3e*, *Figure 3—figure supplements 1c* and *2a*). The two subpopulations also showed the known difference in cycling between large and small pre-BII cells (*Nahar et al., 2011*), with the large cells incorporating 400 times more EdU within 2 hr than the small cells (*Figure 3—figure supplements 1d* and *2b*).

To determine whether the two types of B cell progenitors differ in their plasticity, we isolated them from reprogrammable mice and tested their conversion ability into either macrophages or iPSCs. In response to a continuous exposure to C/EBPa the small pre-BII cells upregulated the macrophage marker Mac-1 faster and downregulated CD19 slightly more rapidly than large pre-BII cells (*Figure 3f–g*, *Figure 3—figure supplement 2c*). Similarly, the small cells acquired higher granularity and a slightly increased volume compared to the large cells, both markers of mature myeloid cells (*Figure 3—figure supplement 1e*). In stark contrast, when 18 hr pulsed cells (also designated Ba' cells (*Di Stefano et al., 2016*; *Di Stefano et al., 2014*)) were tested for reprogramming ability in response to OSKM induction, large pre-BII cells generated 30x times more iPSC colonies than small pre-B cells (*Figure 3h–i*), which, as opposed to large pre-B cells, die out during the reprogramming time course (*Figure 3—figure supplement 1f*).

Previous work testing different times of C/EBPa induction in pre-B cells before OSKM induction showed that an 18 hr treatment elicited a maximal enhancement of the cells' reprogramming efficiency (*Di Stefano et al., 2014*). Longer exposures, driving the cells into a macrophage-like state, decreased the efficiency (*Di Stefano et al., 2014*), raising the possibility that an accelerated transdifferentiation of the small cells towards macrophages moves them out of the time window required

for high responsiveness. If this was the case, a shorter pulse of C/EBPa should increase the reprogramming responsiveness of the small cells. However, when testing the effect of a 6 h C/EBPa pulse we found that the small cells remained highly resistant to reprogramming, exhibiting fewer iPSC colonies than with the 18 hr pulse (*Figure 3—figure supplement 1g*). Taken together, our results indicate that large and small pre-BII cells exhibit intrinsic differences in their cell conversion plasticities.

## Reprogramming susceptibility correlates with high Myc levels in a broad variety of somatic cell types

The observed correlation between high Myc levels and the propensity of pre-B cells for reprogramming into iPSCs could reflect a peculiarity of lymphoid progenitors. We therefore asked whether Myc activity also correlates with the reprogramming efficiency of other somatic cells, examining existing datasets of 9 hematopoietic and 11 non-hematopoietic cell types (*Takahashi and Yamanaka, 2006*; *Eminli et al., 2009*; *Kim et al., 2008*; *Stadtfeld et al., 2008*; *Aasen et al., 2008*). Strikingly, we found that high Myc expression levels in the starting cell type strongly correlate with a high iPSC reprogramming efficiency across all nine different hematopoietic cell types (R = 0.93, p<0.0001, *Figure 3j*), with GMPs and multipotent progenitors (MPPs) exhibiting the highest levels of Myc component and highest reprogramming efficiencies (*Figure 3k*). Furthermore, Myc component levels also correlate with the reprogramming efficiency of various non-hematopoietic cell types (R = 0.66, p=0.0016). These findings show that high Myc expression levels are strongly predictive for the reprogramming susceptibility of a broad variety of somatic cell types.

## Discussion

Here we have described the transdifferentiation and cell reprogramming trajectories of pre-B cells into either macrophages or iPS cells at the single cell level. The observed high frequencies of both cell type conversions are consistent with deterministic processes. However, our experiments also revealed unexpected heterogeneity among cells in the speed and paths by which transcription factors induce transdifferentiation and reprogramming. Our computational analyses made non-trivial predictions about the origins and the consequence of this heterogeneity, predicting an inverse relationship between the ability of cells to either transdifferentiate or to reprogram. These predictions could be experimentally validated, showing the presence of two distinct cell subsets in the starting pre-B cell population, corresponding to previously described large pre-BII cells and small pre-BII cells, into which they normally differentiate. Surprisingly, we found that these two cell types differ in their cell conversion plasticities: while large pre-BII cells efficiently reprogram into iPSCs through a GMP-like cell state but transdifferentiate more slowly into macrophages, small pre-BII cells reprogram much less efficiently into iPSCs but transdifferentiate more rapidly (*Figure 4*).

The finding that cell propensity for transdifferentiation and reprogramming are inversely coupled suggests that the two types of plasticity are intrinsically different. Moreover, the Myc component

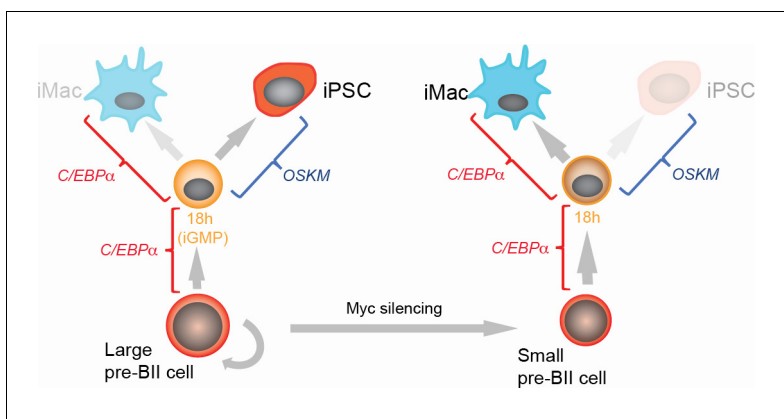

**Figure 4.** Summary of the main findings.
DOI: https://doi.org/10.7554/eLife.41627.014

correlates with both types of plasticity and in a reciprocal manner. Strikingly, high Myc levels correlate with high reprogramming efficiencies not only in hematopoietic progenitors but also in a wide range of other somatic cell types. Consistent with our findings, it has been reported that expression of endogenous *Myc* is essential for efficient reprogramming of MEFs into pluripotent cells (*Hirsch et al., 2015*; *Zviran et al., 2019*). Together, our observations suggest an important role of Myc for the plasticity of both hematopoietic and non-hematopoietic cells. We also discovered that a subset of low Myc component cells at D8 of reprogramming resembles extra-embryonic cell types, reminiscent of an earlier report (*Parenti et al., 2016*).

The Myc effect could be mediated at least in part by one of the different activities reported for the factor, or a combination thereof. These include its ability to induce cell proliferation (*Dang, 2012*), its association with global chromatin changes (*Knoepfler et al., 2006*; *Kieffer-Kwon et al., 2017*), its capacity to transcriptionally activate and amplify genes, including those essential for proliferation (*Lin et al., 2012*) and its induction of metabolic changes (*Dang, 2012*). The unique features of Myc are also likely central for its capacity to act as a major driver of cancer (*Dang, 2012*) and for its role in early embryonic development (*Scognamiglio et al., 2016*). However, high Myc expression in somatic cells is not sufficient to enable their efficient OSKM-induced reprogramming, as we found that large pre-BII cells must still be primed by the transient expression of C/EBPa (*Di Stefano et al., 2014*). This might be related to C/EBPa's multiple functions including its ability to act as a pioneer transcription factor (*van Oevelen et al., 2015*; *Zhu et al., 2018*), to activate key pluripotency TFs such as Klf4, to recruit chromatin related factors including LSD1/Kdm1a, Hdac1, Brd4 and Tet2 (*Di Stefano et al., 2016*; *Sardina et al., 2018*) and/or to induce changes in genome topology preceding pluripotent transcription factor expression (*Stadhouders et al., 2018*). A similar scenario may also play out during OSKM-induced MEF to iPSC reprogramming: here pluripotency factors first target and inactivate enhancers of specific somatic TFs, including *Cebpa*, *Cebpb*, *Fra1* and *Runx1*, before engaging pluripotency gene enhancers (*Chronis et al., 2017*). It therefore appears that efficient reprogramming of somatic into pluripotent stem cells requires three waves of transcription factor activity: i) Expression of high Myc levels in the starting cells, ii) transient expression of specific lineage regulators and iii) activation of key pluripotency transcription factors. It is tempting to speculate that similar transcriptional waves are also required for some of the earliest developmental decisions such as for the formation of pluripotent and extraembryonic cells during pre-implantation embryo development.

## Materials and methods

**Key resources table**

| Reagent type (species) or resource | Designation | Source or reference | Identifiers | Additional information |
|---|---|---|---|---|
| Gene (*Mus musculus*) | cebpa | NA | Ensembl: ENSG00000245848 | |
| Strain, strain background (*Mus musculus*) | Pou5f1GFP transgenic mouse | *Boiani et al., 2002* | NA | Strain: C57Bl/6 × DBA/2 |
| Strain, strain background (*Mus musculus*) | Gt(ROSA)26Sortm1 (rtTA*M2)Jae Col1a1tm3 (tetO-Pou5f1,-Sox2,-Klf4,-Myc)Jae/J | The Jackson Laboratory | Cat# 011004; RRID:IMSR_JAX:011004 | Strain: (C57BL/6 × 129S4/SvJae)F1 |
| Strain, strain background (*Mus musculus*) | Pou5f1-GFP OSKM-reprogrammable | *Jaitin et al. (2014)*, *Di Stefano et al. (2016)* | NA | Strain: C57BL/6 × 129 |
| Cell line (*Homo sapiens*) | PlatE retroviral packaging cell line | Cell Biolabs | Cat# RV-101; RRID: CVCL_B488 | |
| Cell line (*Mus musculus*) | S17 stromal cell line | From Dr. Dorshkind, UCLA. (*Collins and Dorshkind, 1987*) | RRID: CVCL_E226 | |
| Cell line (*Mus musculus*) | Mouse Embryonic Fibroblasts, Irradiated | GIBCO | Cat# A34180 | |

*Continued on next page*

*Continued*

| Reagent type (species) or resource | Designation | Source or reference | Identifiers | Additional information |
|---|---|---|---|---|
| Recombinant DNA reagent | pMSCV-Cebpa-IRES-hCD4 | Produced in-house, (*Bussmann et al., 2009*) | NA | |
| Antibody | Mouse monoclonal APC Anti-human CD4 (RPA-T4) | BD Biosciences | Cat# 555349; RRID: AB_398593 | Dilution used = 1:400 |
| Antibody | Mouse monoclonal biotin anti-human CD4 (RPA-T4) | eBioscience | Cat# 13–0049; RRID:AB_466337 | Dilution used = 1:400 |
| Antibody | Rat monoclonal Anti-Mouse CD16/CD32 (Mouse BD Fc Block) | BD Biosciences | Cat# 553142; RRID: AB_394654 | Dilution used = 1:400 |
| Antibody | Rat monoclonal Pe-cy7 Anti-mouse CD19 (1D3) | BD Biosciences | Cat# 552854; RRID:AB_394495 | Dilution used = 1:400 |
| Antibody | Mouse monoclonal APC Anti-mouse CD11b (44) | BD Biosciences | Cat# 561015; RRID:AB_10561676 | Dilution used = 1:400 |
| Antibody | Rat monoclonal biotin Anti-mouse CD19 (1D3) | BD Biosciences | Cat# 553784; RRID: AB_395048 | Dilution used = 1:400 |
| Antibody | Rabbit monoclonal [Y69] to c-Myc | Abcam | Cat# ab32072; RRID:AB_731658 | Dilution used = 1:76 |
| Antibody | Goat Polyclonal Anti-Rabbit IgG H and L Alexa Fluor 647 | Life technologies | Cat# A32733; RRID:AB_2633282 | Dilution used = 1:2000 |
| Strain, strain background (*Escherichia coli*) | E. coli: BL21(DE3) Competent | New England Biolabs | Cat# C2527I | |
| Peptide, recombinant protein | Recombinant murine IL-7 | Peprotech | Cat# 217–17 | |
| Peptide, recombinant protein | Recombinant murine IL-4 | Peprotech | Cat# 214–14 | |
| Peptide, recombinant protein | Recombinant murine IL-15 | Peprotech | Cat# 210–15 | |
| Peptide, recombinant protein | ESGRO Recombinant mouse LIF protein | Merk Millipore | Cat# ESG1106 | |
| Commercial assay or kit | Click-IT EdU Cytometry assay kit | Invitrogen | Cat# C10425 | |
| Commercial assay or kit | miRNeasy mini kit | Qiagen | Cat# 217004 | |
| Commercial assay or kit | SYBR Green QPCR Master Mix | Applied Biosystems | Cat# 4309155 | |
| Commercial assay or kit | Alkaline Phosphatase Staining Kit II | Stemgent | Cat# 00–0055 | |
| Commercial assay or kit | High Capacity RNA-to-cDNA kit | Applied Biosystems | Cat# 4387406 | |
| Chemical compound, drug | 17β-estradiol | Merck Millipore | Cat# 3301 | |
| Chemical compound, drug | MEK inhibitor (PD0325901) | Selleckchem | Cat# S1036 | |
| Chemical compound, drug | Doxycycline hyclate | Sigma-Aldrich | Cat# D9891 | |

*Continued on next page*

*Continued*

| Reagent type (species) or resource | Designation | Source or reference | Identifiers | Additional information |
|---|---|---|---|---|
| Chemical compound, drug | L-Ascorbic Acid | Sigma-Aldrich | Cat# A92902 | |
| Chemical compound, drug | GSK3b inhibitor (CHIR-99021) | Selleckchem | Cat# S1263 | |
| Other | DMEM Medium | Gibco | Cat# 12491015 | |
| Other | RPMI 1640 Medium | Gibco | Cat# 12633012 | |
| Other | Knockout-DMEM | Gibco | Cat# 10829018 | |
| Other | Neurobasal Medium | Gibco | Cat# 21103049 | |
| Other | DMEM-F12 Medium | Gibco | Cat# 12634010 | |
| Other | Fetal Bovine Serum, E.U.-approved, South America origin | Gibco | Cat# 10270–106 | |
| Other | Embryonic stem-cell FBS, qualified, US origin | Gibco | Cat# 10270–106 | |
| Other | KnockOut Serum Replacement | Gibco | Cat# A3181502 | |
| Other | Pen Strep | Gibco | Cat# 15140122 | |
| Other | L-Glutamine (200 mM) | Gibco | Cat# 25030081 | |
| Other | Sodium Pyruvate (100 mM) | Gibco | Cat# 11360070 | |
| Other | MEM Non-Essential Amino Acids Solution (100X) | Gibco | Cat# 11140068 | |
| Other | 2-Mercaptoethanol | Invitrogen | Cat# 31350010 | |
| Other | N-2 Supplement (100X) | Gibco | Cat# 17502048 | |
| Other | B-27 Serum-Free Supplement (50X) | Gibco | Cat# 17504044 | |
| Other | TrypLE Express Enzyme (1X) | Gibco | Cat# 12605010 | |
| Other | Trypsin-EDTA (0.05%) | Gibco | Cat# 25300054 | |
| Other | MACS Streptavidin MicroBeads | Miltenyi Biotec | Cat# 130-048-101 | |
| Other | MACS LS magnetic columns | Miltenyi Biotec | Cat# 130-042-401 | |
| Software, algorithm | R | R Project for Statistical Computing http://www.r-project.org/ | RRID:SCR_001905 | |

## Mice and cell cultures

We used 'reprogrammable mice' containing a doxycycline-inducible OSKM cassette and the tetracycline transactivator (*Carey et al., 2010*). CD19[+] pre-B cells were isolated from the bone marrow of these mice using monoclonal antibody to CD19 (clone 1D3, BD Pharmingen #553784) and MACS sorting (Miltenyi Biotech). Cell purity was confirmed to be >98% CD19+by FACS using an LSRII machine (BD). After isolation, B cells were grown in RPMI medium supplemented with 10% FBS and 10 ng/ml IL-7 (Peprotech), L-glutamine, nonessential amino acids, β-mercaptoethanol (Life Technologies) as well as penicillin/streptomycin. Mouse embryo fibroblasts (MEFs) were isolated from E13.5 mouse and expanded in DMEM supplemented with 10% FBS, L-glutamine and penicillin/streptomycin. Cultures were routinely tested for mycoplasma contamination. Animal experiments were approved by the Ethics Committee of the Barcelona Biomedical Research Park (PRBB) and performed according to Spanish and European legislation.

## Transdifferentiation and reprogramming experiments

For transdifferentiation pre-B cells were infected with C/EBPαER-hCD4 retrovirus produced by the PlatE retroviral packaging cell line (Cell Biolabs, # RV-101). The cells were expanded for 48 hr on Mitomycin C-inactivated S17 feeders grown in RPMI medium supplemented with 10 ng/mL each of IL-7 (Peprotech) and hCD4$^+$ were sorted (FACSaria, BD). For transdifferentiation C/EBPa was induced by treating the cells with 100 nM β-Estradiol (E2) in medium supplemented with 10 ng/mL each of IL-7, IL-3 (Peprotech) and human colony-stimulating factor 1 (hCSF-1, kind gift of E. Richard Stanley). For reprogramming hCD4$^+$ cells were plated at 500 cells/cm$^2$ in gelatinised plates (12 wells) on irradiated MEF feeders in RPMI medium and pre-treated for 18 hr with E2 to induce C/EBPα. After E2 washout the cultures were switched to serum-free N2B27 medium supplemented with 10 ng/ml IL-4, IL-7 and IL-15 (Peprotech) at 2 ng/ml and treated with 2 µg/ml of doxycycline to activate OSKM. From day two onwards the N2B27medium was supplemented with 20% KSR (Life Technologies), 3 µM CHIR99021 and 1 µM PD0325901 (2i medium). A step-by-step protocol describing the reprogramming procedure can be found at Nature Protocol Exchange (https://www.nature.com/protocolexchange/protocols/4567).

## Myc expression by flow cytometry

CD19 positive B cells were washed and fixed in Fix and Perm fixative (Life Technologies) for 15 min, then washed and permeabilised in Fix and Perm saponin-based permeabilisation buffer for 15 min. After permeabilisation, cells were incubated in 1x PBS/10% normal goat serum/0.3M glycine to block non-specific protein-protein interactions followed by Myc antibody at 1/76 dilution for 30 min at room temperature. The secondary antibody used was Goat Anti-Rabbit IgG H and L (Alexa Fluor 647) (Life technologies) at 1/2000 dilution for 30 min. A rabbit IgG was used as the isotype control. Cells were analysed on a BD LSRII flow cytometer. The gating strategy is described in *Figure 3—figure supplement 2*.

## Cell cycle analysis by EdU incorporation

For cell cycle analyses cells were treated for 2 hr with EdU (Life Technologies). EdU staining was performed using the Click-IT EdU Cytometry assay kit (Life Technologies) at room temperature following the manufacturer's instructions. Briefly, cells were washed in PBS and fixed in Click-iT fixative for 15 min. After washing they were permeabilised in 1 × Click iT saponin-based permeabilisation buffer for 15 min. The EdU reaction cocktail (PBS, CuSO$_4$, Alexa Fluor 488 azide and buffer additive as per manufacturer's protocol) was added to the cells for 30 min and then washed in 1% BSA/PBS. After staining, cells were analysed on a BD LSRII flow cytometer. The gating strategy is described in *Figure 3—figure supplement 2*.

## FACS analyses of transdifferentiation

B cell to macrophage transdifferentiation was monitored by flow cytometry using antibodies against Mac-1 (clone 44, BD Pharmingen) and CD19 (1D3, BD Pharmingen) labelled with APC and PeCy-7, respectively. After staining, cells were analysed on a BD LSRII flow cytometer. The gating strategy is described in *Figure 3—figure supplement 2*.

## RNA extraction

To remove the feeders, cells were trypsinised and pre-plated for 30 min before RNA isolation with the miRNeasy mini kit (Qiagen). RNA was eluted from the columns using RNase-free water and quantified by Nanodrop. cDNA was produced with the High Capacity RNA-to-cDNA kit (Applied Biosystems). qRT-PCR analyses qRT-PCR reactions were set up in triplicate with the SYBR Green QPCR Master Mix (Applied Biosystems). Reactions were run on an AB7900HT PCR machine with 40 cycles of 30 s at 95°C, 30 s at 60°C and 30 s at 72°C.

## Viral vector and infection

Production of the C/EBPαER-hCD4 retroviral vector and B cell infection were performed as before (*Di Stefano et al., 2016*; *Di Stefano et al., 2014*).

## Alkaline Phosphatase (AP) staining

AP staining was performed using the Alkaline Phosphatase Staining Kit (STEMGENT) following the manufacturer's instructions.

## Library preparation and sequencing

Single-cell libraries from polyA-tailed RNA were constructed applying massively parallel single-cell RNA sequencing (MARS-Seq; *Jaitin et al., 2014*) as described in *Guillaumet-Adkins et al. (2017)*. Single cells were FACS isolated into 384-well plates with lysis buffer and reverse-transcription primers containing the single-cell barcodes and unique molecular identifiers (UMIs). Each library consisted of two 192 single-cell pools per time point (pool a and pool b). Multiplexed pools were sequenced in an Illumina HiSeq 2500 system. Primary data analysis was carried out with the standard Illumina pipeline following the manufacturer's protocol.

## Data pre-processing

Quality check of sequenced reads was performed with the FastQC quality control tool (*Andrews, 2010*). Samples that reached the quality standards were then processed to deconvolute the reads to cell level by de-multiplexing according to the pool and the cell barcodes, wherein the first read contains the transcript sequence and the second read the cell barcode and the UMI.

Samples were mapped and gene expression was quantified with default parameters using the RNA pipeline of the GEMTools 1.7.0 suite (*Marco-Sola et al., 2012*) on the mouse genome assembly GRCm38 (*Cunningham et al., 2015*) and Gencode annotations M8 (*Mudge and Harrow, 2015*). We took advantage of the UMI information to correct for amplification biases during the library preparation, collapsing read counts for reads mapping on a gene with the same UMI and considering unambiguously mapped reads only.

## Data analysis

Cells with a library size <1800 were excluded from further analysis. Genes detected in less than 50 cells or less than 15 cells per group were also excluded from further analysis, resulting in expression data for 17183 genes in 3152 cells. Size factor normalisation was applied by dividing the expression of each gene in each cell by the total number of detected mRNA molecules and multiplying by the median number of molecules across cells. An inverse hyperbolic sine transformation (log (x + sqrt ($x^2$+1)), where x is the mRNA count) was then applied and the data were subsequently centred.

## Dimensionality reduction, batch correction and gene expression reconstruction

We performed principal component analysis (PCA) by computing partial singular value decomposition (SVD) on the data matrix extracting the first 100 largest singular values and corresponding vectors using the method implemented in R in the 'irlba' package (*Baglama and Reichel, 2005*). Examining singular vectors highlights the presence of batch effects between the two pools at each time point starting from component 3 (*Figure 1—figure supplement 1c*). We therefore applied a batch correction method based on finding mutual nearest neighbours between batches (*Haghverdi et al., 2017*). We used the R implementation (function 'mnn' in the 'scran' package) with k = 15 nearest neighbours, and computing the nearest neighbours on the first 2 PCA dimensions which only capture biological variation. This method corrects batch effects shared across all samples. However, partial SVD on batch corrected data shows that among the first 35 components that retain signals (*Figure 1—figure supplement 1d*) batch effects between the two pools are still present *Figure 1—figure supplement 1e*). We therefore applied independent component analysis (ICA) to decompose expression into 35 mutually independent components and estimate the relative mixing matrix that, when multiplied by the independent components, results in the observed data (*Figure 1—figure supplement 2d*). ICA separates well sample-specific batch effects from biological signal (*Figure 1—figure supplement 1f*). We filtered out components when the interquartile ranges of the distributions of component scores of the two pools do not overlap at any time point (components 3, 9, 13, 15, 16, 17, 19, 20, 21, 24, 26, 27,32, 35). A component correlated with cell position in the plate (Component 33, *Figure 1—figure supplement 1g*) was also filtered out. We then reconstructed gene expression by multiplying filtered gene loadings (*Supplementary file 1*) by the

filtered samples scores (*Supplementary file 2*) including only the selected 20 components (see *Figure 1—figure supplement 2e* for a schematic description). The resulting gene expression matrix was then normalised using quantile normalisation.

## Computation of similarity index of our single cell RNA-seq data with reference cell types

We compared our data to a comprehensive atlas of murine cell types from *Hutchins et al. (2017)* that consists of uniformly re-analysed bulk and single cell RNA-seq data from 113 publications including 921 biological samples consisting of 272 distinct cell types.

We calculated a similarity score for each single cell transcriptome to each atlas cell type transcriptome as follows: we first calculated the genome wide correlation between each single cell and all cell types from the atlas. The correlation coefficient was then transformed using Fisher's z transformation: $1/2 *ln((1 + r)/(1 r))$. The vector of z-transformed correlations for each single cell was then scaled across reference cell types. In the same manner, we also compared our starting population single cell data to reference bulk expression data from different stages of B cell development from *Hoffmann et al. (2002)* and from the immunological genome project (*Painter et al., 2011*). Myc component increases in expression with time during reprogramming. This may fully account for the prediction of the extent of reprogramming in each cell. We therefore regress out Myc component before the computation of similarity score to derive a corrected similarity index. This was done by reconstructing both the atlas and single cell expression without the Myc component. This shows that Myc component is still well correlated with progression towards pluripotency at least at D4 (*Figure 2—figure supplement 3a*). This holds true when both Myc and cell cycle components are regressed out (*Figure 2—figure supplement 3b*).

## Characterisation of the components: Gene set enrichment analysis

We clustered genes according to the loadings on the components from ICA, assigning each gene to the component with highest or lowest loading. Each component therefore defines one cluster of negatively correlated genes and one of positively correlated genes. We then calculated the enrichment of each cluster for Gene Ontology categories (*Ashburner et al., 2000*), restricting the analysis to categories including more than 10 and less than 200 genes, and hallmark signatures from the Molecular Signature database (*Liberzon et al., 2015*). The hallmark gene set collection consists of 50 refined gene sets derived from over 6700 gene sets of the Molecular Signature Database, which are obtained from a variety of experimental approaches including gene expression profiling and binding location experiments (*Liberzon et al., 2011*). Refinement was obtained by a combination of automated approaches and expert curation, aimed at reducing redundancy among gene sets and expression variation within gene sets (*Liberzon et al., 2015*).

We tested significance of gene set enrichment its significance using Fisher's test. P-values were corrected for multiple testing using Benjamini-Hochberg method (*Benjamini and Hochberg, 1995*).

## Characterisation of the components: comparison to the mouse cell atlas

We compared our data to a comprehensive atlas of murine cell types (*Hutchins et al., 2017*). We applied ICA to decompose expression of the atlas of cell types into 120 mutually independent components, and we correlated these to the components extracted from our single cell data (*Figure 1—figure supplement 2*), to determine cell type specificity of single cell components. To this end, we correlate gene loadings of single cell components with the gene loadings of atlas components. We then defined the cell type specificity of a single cell component as follows: we associate single cell components and atlas components based on the highest absolute value of Pearson's correlation between the gene loadings of the single cell components and of the atlas components (*Figure 1—figure supplement 2a*). For example, the single cell component one gene loadings best correlates with atlas component 12 gene loadings (positive correlation). We next characterise cell type specificity of each atlas component based on the dynamics of the single cell components scores (the single cells' projection onto the components) and on atlas cell type scores (projection of atlas cell type on the atlas component). For example, single cell component one negatively correlates with genes that monotonically increase during transdifferentiation. The correspondent atlas component 12 is

characterised by highly negative scores of macrophage and dendritic cells. We therefore define component 12 of the atlas, and by extension also the single cell component 1, as 'a macrophage' component.

### Diffusion map and diffusion pseudotime

To visualise data in low dimensional space we used diffusion maps. Diffusion maps are a method for non-linear dimension reduction that learn the non-linear data manifold by computing the transition probability of each data point to its neighbours (diffusion distances). We used the R implementation by *Haghverdi et al. (2016)* available in library 'dpt' version 0.6.0. The transition matrix is calculated by using ´Transitions´ function on the selected ICA components using a sigma = 0.12 for the Gaussian kernel. We also calculated diffusion pseudotime using the function 'dpt' in the same library.

### t-SNE

We used the R implementation of t-SNE (Rtsne library). We input the 20 selected components from ICA for the starting pre-B cell population and we choose a perplexity of 30.

### Differential expression analysis, clustering and heatmaps

We performed differential expression analysis on the reconstructed expression using 'limma' package in R, we selected genes differentially expressed at false discovery rate of 5% and with at least 1.3 fold change between adjacent time points during transdifferentiation or reprogramming. We cluster these sets of genes using hierarchical clustering with complete linkage (function hclust in R library 'fastcluster', method='complete'). Clusters are displayed the with heatmaps (function 'heatplot' in 'made4' library). We performed gene set enrichment analyses on these sets of genes and clusters using Fisher's test as explained above.

### Correlation between reprogramming efficiency and myc activity

Reprogramming efficiency data for different hematopoietic cell types as well as from mouse tail fibroblasts are from *Eminli et al. (2009)*; neural stem cells, pancreatic beta cells, keratinocytes and MEFs are from *Kim et al. (2008)*, *Stadtfeld et al. (2008)*, *Aasen et al. (2008)*, and *Takahashi and Yamanaka (2006)*, respectively. Cell reprogramming efficiencies were matched to the expression values of their Myc component, obtained from the mouse cell atlas (*Hutchins et al., 2017*) as described above (*Figure 1—figure supplement 2a,c*). When more than one cell type from the atlas corresponded to a single cell category used for reprogramming, their Myc component values were averaged (*Supplementary file 5*). Myc expression in the hematopoietic lineage is the mean level across single cells of each cell type from *Jaitin et al. (2014)*.

### Data availability

Single cell gene expression data have been deposited in the National Center for Biotechnology Information Gene Expression Omnibus (GEO) under accession number GSE112004.

## Acknowledgements

We would like to thank Ido Amit and Diego Adhemar Jaitin for help with the MARS-Seq technique and the CRG/UPF FACS Unit for help with the cell sorting. Work in the lab of TG was supported by the European Research Council (ERC) Synergy Grant (4D-Genome) and by AGAUR (SGR-1136). Research in the lab of BL was supported by an ERC Consolidator grant (616434), the Spanish Ministry of Economy and Competitiveness (BFU2011-26206), the AXA Research Fund, the Bettencourt Schueller Foundation and AGAUR (SGR-831). We acknowledge support of the Spanish Ministry of Economy and Competitiveness, Centro de Excelencia Severo Ochoa 2013–2017 (SEV-2012–0208) and of the CERCA Programme/Generalitat de Catalunya.

# Additional information

## Funding

| Funder | Grant reference number | Author |
| --- | --- | --- |
| H2020 European Research Council | Synergy Grant (4D-Genome) | Ben Lehner |
| Agency for Management of University and Research Grants | SGR-1136 | Ben Lehner<br>Thomas Graf |
| European Research Council | Consolidator grant (616434) | Ben Lehner |
| Ministry of Economy and Competitiveness | BFU2011-26206 | Ben Lehner |
| AXA Research Fund | | Ben Lehner |
| Fondation Bettencourt Schueller | | Ben Lehner |
| Agency for Management of University and Research Grants | SGR-831 | Ben Lehner |

The funders had no role in study design, data collection and interpretation, or the decision to submit the work for publication.

## Author contributions

Mirko Francesconi, Bruno Di Stefano, Conceptualization, Data curation, Formal analysis, Investigation, Visualization, Methodology, Writing—original draft, Writing—review and editing; Clara Berenguer, Validation, Investigation, Methodology; Luisa de Andrés-Aguayo, Marcos Plana-Carmona, Validation; Maria Mendez-Lago, Validation, Methodology; Amy Guillaumet-Adkins, Marta Gut, Ivo G Gut, Methodology; Gustavo Rodriguez-Esteban, Data curation, Methodology; Holger Heyn, Data curation, Supervision, Methodology, Project administration; Ben Lehner, Conceptualization, Data curation, Supervision, Funding acquisition, Methodology, Writing—original draft, Project administration, Writing—review and editing; Thomas Graf, Conceptualization, Supervision, Funding acquisition, Investigation, Methodology, Writing—original draft, Project administration, Writing—review and editing

## Author ORCIDs

Mirko Francesconi (iD) http://orcid.org/0000-0002-8702-0877
Bruno Di Stefano (iD) http://orcid.org/0000-0003-2532-3087
Marcos Plana-Carmona (iD) https://orcid.org/0000-0002-1976-7506
Ivo G Gut (iD) http://orcid.org/0000-0001-7219-632X
Ben Lehner (iD) https://orcid.org/0000-0002-8817-1124
Thomas Graf (iD) https://orcid.org/0000-0003-2774-4117

## Ethics

Animal experimentation: The protocol was approved by the Committee on the Ethics of Animal Experiments of the Generalitat de Catalunya (Permit Number: JMC-071001P3). All surgery was performed under sodium pentobarbital anesthesia, and every effort was made to minimize suffering.

## Decision letter and Author response

Decision letter https://doi.org/10.7554/eLife.41627.030
Author response https://doi.org/10.7554/eLife.41627.031

## Additional files

### Supplementary files

• Supplementary file 1. Gene cluster membership and gene loadings on each independent component for each detected gene. The sign of cluster membership is positive if the gene has the highest absolute loading on the positive side of the component and negative if the highest absolute loading is on the negative side of the component.
DOI: https://doi.org/10.7554/eLife.41627.015

• Supplementary file 2. Total mRNA count, number of detected genes, and projection onto each independent component, for each single cell.
DOI: https://doi.org/10.7554/eLife.41627.016

• Supplementary file 3. Fisher's test based gene set enrichment analysis on Gene Ontology categories (biological process) for each gene cluster derived from ICA. Includes odds ratios, p-values and FDR, number of genes associated with each category, number and names of genes included both in the cluster and in the category.
DOI: https://doi.org/10.7554/eLife.41627.017

• Supplementary file 4. Fisher's test based gene set enrichment analysis on hallmark genesets for each gene cluster derived from ICA. It includes odds ratio, p-value and FDR, number of genes included in each category, number and names of genes included both in the cluster and in the category.
DOI: https://doi.org/10.7554/eLife.41627.018

• Supplementary file 5. Reprogramming efficiencies for different cell types and expression of Myc from *Jaitin et al. (2014)* and Myc component from the mouse cell type atlas.
DOI: https://doi.org/10.7554/eLife.41627.019

• Supplementary file 6. Fisher's test based gene set enrichment analysis on both GO and hallmark gene sets for genes differentially expressed with a fold change of at least 1.3 between adjacent time points during reprogramming and transdifferentiation. Includes odds ratio, p-value and FDR, number of genes included in each category, number and names of genes both included both in the cluster and in the category.
DOI: https://doi.org/10.7554/eLife.41627.020

• Supplementary file 7. Fisher's test based gene set enrichment analysis on both GO and hallmark gene sets for genes in the clusters shown in the heatmaps of supplementary Figure 3j-l.
DOI: https://doi.org/10.7554/eLife.41627.021

• Transparent reporting form
DOI: https://doi.org/10.7554/eLife.41627.022

### Data availability

Single cell gene expression data have been deposited in the National Center for Biotechnology Information Gene Expression Omnibus (GEO) under accession number GSE112004

The following dataset was generated:

| Author(s) | Year | Dataset title | Dataset URL | Database and Identifier |
|---|---|---|---|---|
| Heyn H, Rodríguez-Esteban G | 2018 | Single cell expression analysis uncouples transdifferentiation and reprogramming | https://www.ncbi.nlm.nih.gov/geo/query/acc.cgi?acc=GSE112004 | NCBI Gene Expression Omnibus, GSE112004 |

The following previously published datasets were used:

| Author(s) | Year | Dataset title | Dataset URL | Database and Identifier |
|---|---|---|---|---|
| Hoffmann R, Seidl T, Neeb M, Rolink A, Melchers F, Rolink T | 2002 | Murine bone marrow B cell precursors | https://www.ncbi.nlm.nih.gov/geo/query/acc.cgi?acc=GSE13 | NCBI Gene Expression Omnibus, GSE13 |
| The Immunological | 2009 | Immunological Genome Project | http://www.ncbi.nlm.nih. | NCBI Gene |

| Genome Project Consortium | data Phase 1 | gov/geo/query/acc.cgi?acc=GSE15907 | Expression Omnibus, GSE15907 |

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
