## [Decision Letter]

Thank you for submitting your article "Single cell expression analysis uncouples transdifferentiation and reprogramming" for consideration by *eLife*. Your article has been reviewed by three peer reviewers, including Chris P Ponting as the Reviewing Editor and Reviewer #1, and the evaluation has been overseen by Michel Nussenzweig as the Senior Editor.

The reviewers have discussed the reviews with one another and the Reviewing Editor has drafted this decision to help you prepare a revised submission.

Summary:

Francesconi et al. have undertaken a time series experiment to compare pre-B cell transdifferentiation into macrophages, with reprogramming into iPSCs. Using single cell RNA sequencing they identify the sequential loss of B cell specific factors and the engagement of myeloid gene expression patterns during transdifferentiation. Concordantly they show that reprogramming leads to a loss of B cell identity and a gain of ESC-like transcriptional state 8 days after reprogramming. Further, they correlate these changes with a number of different factors and ultimately show that expression of Myc, and its downstream target genes, in the starting population are predictive of transdifferentiation and reprogramming efficiency. This difference is predicted by Myc expression but Myc was not demonstrated unequivocally to be causal. This model is consistent with functional experiments, which reveal that the initial starting population was heterogeneous, and consisted of 2 distinct B cell developmental stages, each with reciprocal efficiencies for transdifferentiation and reprogramming.

Essential revisions:

1) Some decisions made in how the single cell data are processed to be poorly motivated and somewhat arbitrary. Additionally, it is not clear how the approach and experimental design taken by the authors address their specific problem, and why a simpler approach could not be taken. In particular, and rather surprisingly, the authors do not check for heterogeneity in the starting population, for instance, is there any indication of structure in the pre-stimulated cells? This could be interrogated very simplistically, particularly as the authors make pains to point out bi- and tri-modality in hard to read violin plots for latter stages post-stimulation. Subsection “Large and small pre-B cells differ reciprocally in their propensity to transdifferentiate or reprogram”, last paragraph, could the small and large pre-B cells be in fundamentally different states?

In the Discussion it is said that "a cell's propensity for transdifferentiation or reprogramming can be disassociated, indicating that these two types of plasticity are fundamentally different". This is overstating the evidence because this statement discusses "a cell" whereas the propensity differences are evident for two different cell types (albeit one differentiation step apart). The authors appear not to show, using highly purified populations of these 2 cell types, that "these two types of plasticity are fundamentally different."

OSKM includes MYC. Why does input MYC matter? Is it merely a marker for the two different cell types that differ in some other way? This needs to be clarified.

The conclusion that MYC level inhibits lineage conversion is based on a single model, which only uses one transcription factor. A general statement should not be made based only on a single model.

Analysis of open chromatin across the time points could be helpful. Does CEBPa open new chromatin or only bind to existing open chromatin sites?

2) The reviewers agree that it is plausible that MYC expression level affects reprogramming. The evidence for this is, however, correlative: the mechanism by which MYC elicits its effects is not determined. To be definitive, the authors could demonstrate that low levels of endogenous Myc (via CRISPRi and/or knockdown) are associated with inefficient reprogramming in one (or preferably more) cell types. It may also be true that cycling cells are easier to reprogram. This may not represent a novel discovery (Tsubouchi et al., Cell 2013). Proliferating cells dilute out existing proteins when transcription rates drop, this may be needed for loss of the starting fate. More generally, at several points the authors state that "X predicts Y". It would be more accurate to state that "X is correlated with Y" since prediction implies a causality that is not proved.

3) The Results section is hard-going at times, particularly when the reader is expected to rederive the conclusions of the text simultaneously using many different figures and panels. Examples are:a) "We found that a signature enriched in Myc target genes (component 5, Figure 1—figure supplement 2, Supplementary file 4) best predicts speed of cell conversion and negatively correlates with the progression of cells at intermediate time points of transdifferentiation (Figure 2B, Figure 2—figure supplement 1)." Here, the reader is requested to compare 3 figures (and 1 table) whilst figuring out what is meant by "speed of cell conversion". We request that you take your time to explain these observations so that the reader better appreciates this important sentence. re: Figure 2D. "low expression of Myc targets is more strongly associated with a rapid gain of the macrophage state than with a rapid loss of the B cell state". Again there is a need to lead the reader through this plot.

4) The authors will need to introduce the experimental system in more detail, in particular the dox-inducible cassette encoding the 4 Yanamaka factors. Additionally, they should provide some justification for the different time points and resolution between the transdifferentiation and reprogramming experiments.

5) When describing the results of the single cell experiments, there needs to be more specific examples of genes and their relevance. Simply describing a 'B cell program' is uninformative. To that end, the presentation of the top 50% of expressed cells feels a little like cherry picking; the authors should present the distribution of expression for all cells, not just the top 50%.

6) Related to the above, the last paragraph of the subsection “Single cell analysis of highly efficient transdifferentiation and reprogramming from the same cell population”, should be split into 2 separate sections, and expanded to include more details as above.

7) The subsection “Rapid transdifferentiation into macrophages is associated with low expression of Myc targets” could potentially be explained by differences in resolution of the time series experiments. The authors need to demonstrate quantitatively that there are differences in the rates of transdifferentiation versus reprogramming, and what are the potential explanations. This is not aided by the difficulty in reading many of the plots, i.e. violin plots, then stating something is evident when it has not been demonstrated. The presentation of these data needs to be clearer. It is better to provide several clear examples that illustrate the authors' point, than to try to squeeze too many plots into a single figure panel.

8) Subsection “The rapid transdifferentiation of cells with low Myc activity is more direct” – it is not clear that higher Myc target expression is associated with a more persistent induction of a GMP-like state, as there is no indication of a relative time difference between Myc-high and Myc-low cells.

9) Unsurprisingly the monocyte and macrophage components are highly correlated – given this Figure 2F is arguably redundant.

10) Subsection “Efficient reprogramming is associated with high expression of Myc targets” – please confirm that the ICM component is not driven by Myc target genes; include a reference to Figure 2—figure supplement 3 for this.

11) Would any population of cycling unsynchronized cells also show considerable variation in Myc target gene expression? A proper comparison to control populations that are synchronized or unsynchronized, e.g. ESCs, would give some indication of where these pre-B cells sit in this respect.

12) Subsection “Variation in Myc activity reflects pre-existing variation in the starting cell population”, last sentence. This statement is too strong. Specifically, at no point have the authors presented evidence about speeds or rates of transdifferentiation and/or reprogramming.

13) Figure 3D – can you justify the thresholds used for the FSC and SSC? In particular, in later figure panels it seems like there is a strong overlap between the "large" cells and "small" cells.

14) a) From a more technical perspective, what is the motivation of using an inverse hyperbolic sine transformation, when a scaling factor would achieve the same result? If it does not, then demonstrate why.

b) Additionally, correction with MNN removes batch effects that are orthogonal to the biological signal. Using only first 2 PCs will likely miss genuine MNNs, and thus will not appropriately correct batch effects embedded in deeper reduced dimensions.

c) Subsection “Computation of similarity index of our single cell RNA-seq data with reference cell types”. Was this regressed out of the similarity index, or were the Myc target genes just removed before re-computing the scores? This needs clarification. More generally, a gene is expressed, not a signature, i.e. talking about the expression of a Myc signature/component does not make biological sense – it is an artificial construction. Say score, or loading, instead of expression.

d) Subsection “Characterization of the components: comparison to the mouse cell atlas” – what correlation coefficient was used here? Pearson's, Spearman's, Kendall's, etc.?

e) Subsection “Diffusion map and diffusion pseudotime” – why use a sigma of 0.12, this seems very specific?

f) Subsection “Differential expression analysis, clustering and heatmaps” – when describing a differential expression analysis it is standard practise to indicate the false discovery rate. Also, why use a fold change of 1.3? Again, this is a strangely arbitrary value that needs some justification.

15) Other issues:

a) Figure 1—figure supplement 3 caption is not complete – sub panel indicators are missing.

b) Abstract. Make it clearer that the two cell types are categorically different ("only a single differentiation step apart", Discussion, also shown in Figure 4).

c) Figure 1I. Clarify why the 6h data were not included.

d) Figure 1—figure supplement 2A. Justify the selections of components and cell types. Explain "Scheme of ICA decomposition the"

e) Figure 1—figure supplement 4 (subsection “Rapid transdifferentiation into macrophages is associated with low expression of Myc targets”) "This is also evident from the bi- or trimodal distribution of key marker genes at intermediate time-points of reprogramming and transdifferentiation". The bi/trimodality is not immediately evident from these violin plots so I suggest that you show the raw data for an exemplar marker gene and cell type (e.g. Nanog). In the text this is described as "asynchronous behaviour" yet formally this is an interpretation and at this stage in the manuscript no data supporting this has been provided.

f) Figure 2B. Explain that the start/end point data are not shown, and say why in the legend.

g) Explain at greater length the "time window of highest responsiveness".

h) Introduction, "homogeneous".

i) Subsection “Efficient reprogramming is associated with high expression of Myc targets”, change "can" to "could".

j) Explain (re: 0.01% to 25%) that the conversion processes remain unclear.

---

## [Author Response]

Essential revisions:1) Some decisions made in how the single cell data are processed to be poorly motivated and somewhat arbitrary. Additionally, it is not clear how the approach and experimental design taken by the authors address their specific problem, and why a simpler approach could not be taken.

In the revised manuscript we have both better explained the motivations behind the data processing and, as detailed in the responses to individual queries below, we have also included analyses to show that alternative processing of the data give rise to the same conclusions.

The experimental design is also better introduced in the Introduction. The aim was to address how heterogeneous these ‘efficient’ transdifferentiation and reprogramming systems really are. Single cell RNA sequencing is the obvious unbiased approach to address this question.

In particular, and rather surprisingly, the authors do not check for heterogeneity in the starting population, for instance, is there any indication of structure in the pre-stimulated cells? This could be interrogated very simplistically, particularly as the authors make pains to point out bi- and tri-modality in hard to read violin plots for latter stages post-stimulation.

When we initiated our experiments, we had no indication that the isolated pre-B cells are heterogeneous. Rather, this was a bold prediction that we made based on computational analyses of the single cell RNA-seq time series. We could clearly see that there was heterogeneity in the trans-differentiating and reprogramming cells. A major part of this heterogeneity related to expression of Myc target genes and this also related to variation in the total RNA content of each single cell. This could either have been heterogeneity induced by the TF pulse or heterogeneity that pre-existed in the starting cell population such that cells responded differently to the pulse. We tested this hypothesis and found that indeed the starting cell population consists of two discrete populations of cells with different sizes and cell cycle states. We show that these two populations differ substantially and reciprocally in their efficiency of trans-differentiation and reprogramming. In short, we think this study represents a great example of how unbiased single cell expression profiling can lead to completely unexpected (to us) and fundamental insights into a system that we have been studying for a very long time.

We have now included a two-dimensional t-SNE plot (Figure 3—figure supplement 1A) to show that the structure in the starting pre-B cell population largely overlaps with variation in the Myc component.

Subsection “Large and small pre-B cells differ reciprocally in their propensity to transdifferentiate or reprogram”, last paragraph, could the small and large pre-B cells be in fundamentally different states?

Small and large pre-BII cells indeed represent different cell states: large pre-BII cells have higher Myc expression and activity and are proliferating. In contrast, small pre-BII cells are Myc low and quiescent. This is shown in Figure 3A-E, Figure 3—figure supplement 1A-D, and described in the Results section.

In the Discussion it is said that "a cell's propensity for transdifferentiation or reprogramming can be disassociated, indicating that these two types of plasticity are fundamentally different". This is overstating the evidence because this statement discusses "a cell" whereas the propensity differences are evident for two different cell types (albeit one differentiation step apart). The authors appear not to show, using highly purified populations of these 2 cell types, that "these two types of plasticity are fundamentally different."

We show that purified small pre-BII cells trans-differentiate faster to become macrophages (Figure 3F, G) and are almost refractory to reprogramming into induced pluripotent stem cells. In contrast, large pre-BII cells (Figure 3H, I) trans-differentiate more slowly (and via a different path) but reprogram very efficiently. It is the comparison between these two very related cell types that we are making. Naively one might think that a more ‘plastic’ cell would respond more efficiently to both transitions. In contrast, what we see is that the cell type that reprograms better trans-differentiates worse and vice versa. We have tried to explain this better in the revised manuscript and have removed the term ‘fundamentally’.

OSKM includes MYC. Why does input MYC matter? Is it merely a marker for the two different cell types that differ in some other way? This needs to be clarified.

At the moment we cannot distinguish whether Myc activity is a marker that predicts the ability to reprogram from whether Myc is causally important in driving these differences between cell types. However, the fact that Myc levels predict the ability to reprogram across many different cell types (Figures 3J and K) and the known role of Myc in reprogramming make it a parsimonious explanation that Myc activity is causal. Based on the literature the effect of Myc expression in the starting cells could be mediated by at least three different scenarios as we now discuss in more detail:

“The Myc effect could be mediated by either one of different activities reported for the factor, or a combination of them. These include its ability to induce cell proliferation (Dang, 2012), its association with global chromatin changes (Knoepfler et al., 2006; Kieffer-Kwon et al., 2017), its capacity to transcriptionally amplify a large number of genes, including those essential for proliferation (Lin et al., 2012) and its induction of metabolic changes (Dang, 2012).”

The conclusion that MYC level inhibits lineage conversion is based on a single model, which only uses one transcription factor. A general statement should not be made based only on a single model.

We agree and have modified the text appropriately (subsection “Variation in Myc activity reflects pre-existing variation in the starting cell population”, first paragraph). Myc levels predict reprogramming efficiency across a very wide range of cell types. However, we have only shown that Myc levels inhibit the ability to trans-differentiate for a single cell fate change.

Analysis of open chromatin across the time points could be helpful. Does CEBPa open new chromatin or only bind to existing open chromatin sites?

As we have described earlier^6,7^, C/EBPa can bind not only to sites primed by other transcription factors, but also to closed chromatin, acting as a pioneer factor. For example, during iPS reprogramming C/EBPa primes pluripotency-associated enhancer regions for subsequent binding of the transcription factor Klf4^7^. Recent work by the Taipale lab has extended these observations^8^. This is now mentioned in the text (subsection “Large and small pre-B cells differ reciprocally in their propensity to transdifferentiate or reprogramrespective transdifferentiation and reprogramming propensities”, second paragraph).

2) The reviewers agree that it is plausible that MYC expression level affects reprogramming. The evidence for this is, however, correlative: the mechanism by which MYC elicits its effects is not determined. To be definitive, the authors could demonstrate that low levels of endogenous Myc (via CRISPRi and/or knockdown) are associated with inefficient reprogramming in one (or preferably more) cell types.

While our manuscript was under review it has been reported that the knock-out of Myc in fibroblasts prior to OSKM overexpression significantly reduces iPSC reprogramming efficiency^1^, corroborating our findings. This paper is now cited in the text.

It may also be true that cycling cells are easier to reprogram. This may not represent a novel discovery (Tsubouchi et al., Cell 2013). Proliferating cells dilute out existing proteins when transcription rates drop, this may be needed for loss of the starting fate. More generally, at several points the authors state that "X predicts Y". It would be more accurate to state that "X is correlated with Y" since prediction implies a causality that is not proved.

This is an interesting suggestion but our data do not support the dilution model because the Myc component correlates more strongly with the gain of the macrophage or pluripotency program than the loss of B cell program (Figure 2D and J).

We have now toned down our wording and state that Myc correlates with, rather than predicts susceptibility to reprogramming into pluripotent cells.

3) The Results section is hard-going at times, particularly when the reader is expected to rederive the conclusions of the text simultaneously using many different figures and panels. Examples are:a) "We found that a signature enriched in Myc target genes (component 5, Figure 1—figure supplement 2, Supplementary file 4) best predicts speed of cell conversion and negatively correlates with the progression of cells at intermediate time points of transdifferentiation (Figure 2B, Figure 2—figure supplement 1)." Here, the reader is requested to compare 3 figures (and 1 table) whilst figuring out what is meant by "speed of cell conversion". We request that you take your time to explain these observations so that the reader better appreciates this important sentence. re: Figure 2D. "low expression of Myc targets is more strongly associated with a rapid gain of the macrophage state than with a rapid loss of the B cell state". Again there is a need to lead the reader through this plot.

We have now extensively rewritten the Results section to make it easier to read.

4) The authors will need to introduce the experimental system in more detail, in particular the dox-inducible cassette encoding the 4 Yanamaka factors. Additionally, they should provide some justification for the different time points and resolution between the transdifferentiation and reprogramming experiments.

We have now added a description of the experimental model used in the manuscript (see Introduction).

The time points chosen for the induced cell reprogramming correspond to those used in earlier publications^7,9,10^. Those for transdifferentiation are essentially as described earlier^11-13^, except that we added the 18hrs time point to permit a comparison with the reprogramming protocol. Thus, the time-points described in Figure 1A permit a direct comparison with the studies on bulk populations performed earlier. This has now been explained in the subsection “Single cell analysis of highly efficient transdifferentiation and reprogramming from the same cell population”.

5) When describing the results of the single cell experiments, there needs to be more specific examples of genes and their relevance. Simply describing a 'B cell program' is uninformative. To that end, the presentation of the top 50% of expressed cells feels a little like cherry picking; the authors should present the distribution of expression for all cells, not just the top 50%.

We have now explained the functions of some of the B cell and myeloid genes shown as examples in Figure 1 and Figure 1—figure supplement 3C (see subsection “C/EBPa expression silences the B cell program and induces a progenitor state followed by a monocyte/macrophage program”).

Moreover, we have shown the distribution across all cells for all the genes in the different programs regulated during reprogramming and transdifferentiation in the heatmaps in Figure 1—figure supplement 3J, K and L. We also performed a functional gene set analysis of each cluster highlighted in the heatmaps and presented this analysis in Supplementary file 6 and 7. These tables also presents the genes in each cluster that are also included in each gene set.

6) Related to the above, the last paragraph of the subsection “Single cell analysis of highly efficient transdifferentiation and reprogramming from the same cell population”, should be split into 2 separate sections, and expanded to include more details as above.

We have now introduced a new subheading entitled:

“Cell conversion trajectories suggest deterministic nature of transcription factor induced transdifferentiation and reprogramming”.

We have now stated: “Our findings therefore support the notion that both transcription factor-induced transdifferentiation and reprogramming represent deterministic processes”.

We have now introduced a new subheading entitled:

“C/EBPa expression silences the B cell program and induces a progenitor state followed by a monocyte/macrophage program”.

We have now introduced a new subheading entitled:

“OSKM expression in C/EBPa-pulsed cells further accelerates B cell silencing and leads to the sequential upregulation of the pluripotency program”.

These last two sections were also expanded to include the description of relevant genes whose expression changes plus a short conclusion.

7) The subsection “Rapid transdifferentiation into macrophages is associated with low expression of Myc targets” could potentially be explained by differences in resolution of the time series experiments. The authors need to demonstrate quantitatively that there are differences in the rates of transdifferentiation versus reprogramming, and what are the potential explanations. This is not aided by the difficulty in reading many of the plots, i.e. violin plots, then stating something is evident when it has not been demonstrated. The presentation of these data needs to be clearer. It is better to provide several clear examples that illustrate the authors' point, than to try to squeeze too many plots into a single figure panel.

The reviewer might have misunderstood that we were trying to compare the rate/speed of reprogramming and transdifferentiation. However, what we were interested in was whether there are differences in timing among single cells within each cell fate transition. Indeed, the diffusion map in Figure 1B and even more clearly, the projection of single cells onto the B cell-, monocyte- and macrophage- specific components in Figure 1H, show that cells at 42 hours after C/EBPa induction are spread into three different clusters along the transdifferentiation trajectory: some cells are similar to those at earlier time points, others are similar to later time points and some are at an intermediate stage. This variability at intermediate time points is also visible in Figure 2A which shows the similarity of single cell transcriptome to reference macrophage transcriptome, and in Figure 1—figure supplement 4 where we substituted violin plots of single markers with dot plots, as suggested. This heterogeneity indicates that cells do not undergo transdifferentiation at the same speed but rather exhibit an asynchronous behaviour. Similar observations were made for the reprogramming trajectory (subsections “Heterogeneity at intermediate time pointssuggests asynchrony in transdifferentiation timing”, “Rapid transdifferentiation into macrophages is associated with low expression of Myc targets” and “Efficient reprogramming correlates with high Myc targets expression”, last paragraph).

8) Subsection “The rapid transdifferentiation of cells with low Myc activity is more direct” – it is not clear that higher Myc target expression is associated with a more persistent induction of a GMP-like state, as there is no indication of a relative time difference between Myc-high and Myc-low cells.

We respectfully disagree. From Figure 2E it is evident that the high Myc cells (green/blue) resemble GMPs up to 42hrs after C/EBPa induction, whereas the low Myc cells (yellow/orange) only show some similarity to GMPs at 18hrs after induction. In the subsection “Cells with high Myc activity transdifferentiate via a pronounced GMP-like cell state”, we have now reworded the text to make this clearer.

9) Unsurprisingly the monocyte and macrophage components are highly correlated – given this Figure 2F is arguably redundant.

We have now removed Figure 2F and substituted it with the former Supplementary Figure 6.

10) Subsection “Efficient reprogramming is associated with high expression of Myc targets” – please confirm that the ICM component is not driven by Myc target genes; include a reference to Figure 2—figure supplement 3 for this.

The conclusions about similarity to ICM hold even when the Myc component is regressed out of both our single cell and the cell atlas gene expression data before calculating the similarity score. This is shown in Figure 2—figure supplement 3. We have now also added the corresponding reference relevant for this figure in the second paragraph of the subsection “Efficient reprogramming correlates with high Myc targets expression”.

11) Would any population of cycling unsynchronized cells also show considerable variation in Myc target gene expression? A proper comparison to control populations that are synchronized or unsynchronized, e.g. ESCs, would give some indication of where these pre-B cells sit in this respect.

We don’t fully understand this point. If the reviewer suspects that the Myc component simply captures variations in cell cycle in our single cell data, we can assert that although correlated, Myc activity and cell cycle are at least partially independent. Thus, our ICA analysis distinguishes Myc targets from G2/M and S phase components (see Author response image 1). Further, applying ICA to the cell atlas, which is based on bulk data obtained from unsynchronized cells, also reveals a Myc component distinct from the cell cycle component, which here includes both G2/M and S components.

Moreover, the effect of Myc expression on protein synthesis and cells size has also been experimentally shown to be independent of the cell cycle at all stages of B lymphocyte development^14^. In short, the Myc and cell cycle components are separable.

12) Subsection “Variation in Myc activity reflects pre-existing variation in the starting cell population”, last sentence. This statement is too strong. Specifically, at no point have the authors presented evidence about speeds or rates of transdifferentiation and/or reprogramming.

We agree with the reviewer and have now changed the sentence into the following:

“Taken together, our analyses suggest a pre-existing heterogeneity in the starting cell population, corresponding to large and small pre-BII cells. Moreover, they suggest that small pre-BII cells should transdifferentiate faster but reprogram more slowly, while large pre-BII cells should transdifferentiate more slowly but reprogram faster”.

13) Figure 3D – can you justify the thresholds used for the FSC and SSC? In particular, in later figure panels it seems like there is a strong overlap between the "large" cells and "small" cells.

We assume that the reviewer refers to former Supplementary Figure 9 (current Figure 3—figure supplement 1). We agree that in these experiments the distinction between large and small cells is not entirely clear. For this reason we have introduced three gates: one for small cells with low granularity, one for large, granular cells and one for intermediate sized cells serving as a ‘buffer’. The results obtained for Myc expression justify this admittedly somewhat arbitrary separation.

14) a) From a more technical perspective, what is the motivation of using an inverse hyperbolic sine transformation, when a scaling factor would achieve the same result? If it does not, then demonstrate why.

Inverse hyperbolic sine transformation, defined as log(x+(χ2+1)^1/2^), is a log-like transformation commonly used in RNA-seq analyses. It has the advantage over logarithmic transformation that it is defined for 0 counts. It tends to log(2x) for large x and to log(1) = 0 for small x. It is therefore equivalent to log (x +1), which is also a commonly used transformation for RNA-seq data. This transformation makes the distribution of counts closer to a normal distribution although distributions among single cells are not comparable. The scaling factor on the other hand serves the purpose of making the distribution of expression values comparable among cells despite different number of total reads. Therefore, applying a scaling factor (which we also do) and transforming the data using inverse hyperbolic sine transformation are two distinct but not mutually exclusive procedures that serve different purposes.

b) Additionally, correction with MNN removes batch effects that are orthogonal to the biological signal. Using only first 2 PCs will likely miss genuine MNNs, and thus will not appropriately correct batch effects embedded in deeper reduced dimensions.

Some batch effects embedded in reduced dimensions might be missed at this step but we further removed any residual batch effects by applying ICA extracting 35 components (which capture all the meaningful variance in the data) and filtering out 15 independent components showing residual batch effects, in particular time point specific batch effects which were not captured by MNN. This is explained in Materials and methods (subsection “Dimensionality reduction, batch correction and gene expression reconstruction”) and illustrated in Figure 1—figure supplement 1.

Moreover, we have tried to use MNN batch correction using the first 35 PCs but after applying ICA and extracting 35 independent components we still find components capturing batch effects, including time point-specific batch effects (see for example component 9 and component 24 in Author response image 2).

**Author response image 2. respfig2:** 

c) Subsection “Computation of similarity index of our single cell RNA-seq data with reference cell types”. Was this regressed out of the similarity index, or were the Myc target genes just removed before re-computing the scores? This needs clarification.

The Myc component was regressed out of the expression data before re-computing the similarity scores. This is now explained better in the Materials and methods section and reads as follows.

“We therefore regress out the Myc component before computation of the similarity index to derive a corrected similarity index. This was done by reconstructing both the cell atlas and our single cell expression data without the Myc component”.

More generally, a gene is expressed, not a signature, i.e. talking about the expression of a Myc signature/component does not make biological sense – it is an artificial construction. Say score, or loading, instead of expression.

We now changed this into Myc target gene expression.

d) Subsection “Characterization of the components: comparison to the mouse cell atlas” – what correlation coefficient was used here? Pearson's, Spearman's, Kendall's, etc.?

It’s a Pearson’s correlation. This is now specified.

e) Subsection “Diffusion map and diffusion pseudotime” – why use a sigma of 0.12, this seems very specific?

The diffusion map is just for visualization purposes. We used the ‘find_sigmas’ function in the R library ‘destiny’ to automatically select the sigma of the Gaussian kernel. This function suggested a sigma of 0.1. With this value however, the trajectories look very compressed and difficult to visualize. That is the only reason we chose a slightly larger sigma of 0.12. We have now included the sigma 0.1 and sigma 0.15 versions to show the effect of varying sigma in Figure 1—figure supplement 2F. As can be appreciated, the main message of the figure does not change if we use sigma = 0.1, 0.12 or 0.15 (Author response image 3).

**Author response image 3. respfig3:** 

f) Subsection “Differential expression analysis, clustering and heatmaps” – when describing a differential expression analysis it is standard practise to indicate the false discovery rate. Also, why use a fold change of 1.3? Again, this is a strangely arbitrary value that needs some justification.

All genes presented in the heatmaps are differentially expressed at 5% FDR. However, with so many cells even small changes in gene expression are significant at 5% FDR. Thus, we added an additional fold change cut off to reduce the number of genes visualized in the heatmaps. We agree this cut off is arbitrary, however it is as arbitrary as any other cut off on fold change or FDR. We based our conclusions on gene set enrichment analysis, which is robust against the inclusion or exclusion of single genes close to the threshold. Repeating the analysis with more or less stringent cut off on the fold change does not alter the conclusions on the processes that are regulated during transdifferentiation and reprogramming. For example, 126 genes are upregulated with a fold change of 1.3 between 6h and 18h after C/EBPa induction. If we further restrict the cut off to 1.4, the number of genes is reduced to 67 and if we use a cut off of 1.2 it increases to 288. However, the gene set enriched categories that are over represented among these genes are basically the same (innate immune response, cytokine production, inflammation, see attached excel table), showing that our analysis is robust even when changing thresholds.

15) Other issues:a) Figure 1—figure supplement 3 caption is not complete – sub panel indicators are missing.

We have now corrected this.

b) Abstract. Make it clearer that the two cell types are categorically different ("only a single differentiation step apart", Discussion, also shown in Figure 4).

We have now simplified and modified the Abstract accordingly

c) Figure 1I. Clarify why the 6h data were not included.

We have now included this time point.

d) Figure 1—figure supplement 2A. Justify the selections of components and cell types. Explain "Scheme of ICA decomposition the"e) Figure 1—figure supplement 4 (subsection “Rapid transdifferentiation into macrophages is associated with low expression of Myc targets”)"This is also evident from the bi- or trimodal distribution of key marker genes at intermediate time-points of reprogramming and transdifferentiation". The bi/trimodality is not immediately evident from these violin plots so I suggest that you show the raw data for an exemplar marker gene and cell type (e.g. Nanog).

We now better explain the selection of components and cell types in the Materials and methods section. It now reads as follows:

“Characterization of the components: comparison to the mouse cell atlas. We compared our data to a comprehensive atlas of murine cell types (Hutchins et al., 2017). […] We then defined the cell type specificity of a single cell component based on the cells that have the most extreme distribution in the sample scores of the atlas component(s) with the highest absolute Pearson’s correlation values with the single cell component.” We also now show dot plots instead of violin plots in Figure 1—figure supplement 4.”

In the text this is described as "asynchronous behaviour" yet formally this is an interpretation and at this stage in the manuscript no data supporting this has been provided.

We have now modified this into: “apparent asynchronous behaviour”.

f) Figure 2B. Explain that the start/end point data are not shown, and say why in the legend.

We now included a sentence in the legend clarifying this (Figure 2B legend).

g) Explain at greater length the "time window of highest responsiveness".

We have now added a sentence to explain what is meant by this, referring to our earlier work: “Previous work testing different times of C/EBPa induction in pre-B cells before OSKM induction showed that an 18h treatment elicited a maximal enhancement of the cells’ reprogramming efficiency. Longer exposures, driving the cells into a macrophage-like state, decreased the efficiency^9^, raising the possibility that an accelerated transdifferentiation of the small cells towards macrophages moves them out of the time window required for high responsiveness.”

h) Introduction, "homogeneous".

Done.

i) Subsection “Efficient reprogramming is associated with high expression of Myc targets”, change "can" to "could".

We corrected this.

j) Explain (re: 0.01% to 25%) that the conversion processes remain unclear.

We have now added: varies widely, ranging between 0.01% for T lymphocytes to 25% for GMPs for unclear reasons.

References:

1) Zviran, A. et al. Deterministic Somatic Cell Reprogramming Involves Continuous Transcriptional Changes Governed by Myc and Epigenetic-Driven Modules. Cell Stem Cell, doi:10.1016/j.stem.2018.11.014 (2018).

2) Kieffer-Kwon, K. R. et al. Myc Regulates Chromatin Decompaction and Nuclear Architecture during B Cell Activation. Mol Cell 67, 566-578 e510, doi:10.1016/j.molcel.2017.07.013 (2017).

3) Prieto, J. et al. MYC Induces a Hybrid Energetics Program Early in Cell Reprogramming. Stem Cell Reports 11, 1479-1492, doi:10.1016/j.stemcr.2018.10.018 (2018).

4) Palmieri, S., Kahn, P. and Graf, T. Quail embryo fibroblasts transformed by four v-myc-containing virus isolates show enhanced proliferation but are non tumorigenic. EMBO J 2, 2385-2389 (1983).

5) Tsubouchi, T. et al. DNA synthesis is required for reprogramming mediated by stem cell fusion. Cell 152, 873-883, doi:10.1016/j.cell.2013.01.012 (2013).

6) van Oevelen, C. et al. C/EBPalpha Activates Pre-existing and de novo Macrophage Enhancers during Induced Pre-B Cell Transdifferentiation and Myelopoiesis. Stem Cell Reports 5, 232-247, doi:10.1016/j.stemcr.2015.06.007 (2015).

7) Di Stefano, B. et al. C/EBPalpha creates elite cells for iPSC reprogramming by upregulating Klf4 and increasing the levels of Lsd1 and Brd4. Nat Cell Biol 18, 371-381, doi:10.1038/ncb3326 (2016).

8) Zhu, F. et al. The interaction landscape between transcription factors and the nucleosome. Nature 562, 76-81, doi:10.1038/s41586-018-0549-5 (2018).

9) Di Stefano, B. et al. C/EBPalpha poises B cells for rapid reprogramming into induced pluripotent stem cells. Nature 506, 235-239, doi:10.1038/nature12885 (2014).

10) Stadhouders, R. et al. Transcription factors orchestrate dynamic interplay between genome topology and gene regulation during cell reprogramming. Nat Genet 50, 238-249, doi:10.1038/s41588-017-0030-7 (2018).

11) Xie, H., Ye, M., Feng, R. and Graf, T. Stepwise reprogramming of B cells into macrophages. Cell 117, 663-676 (2004).

12) Bussmann, L. H. et al. A robust and highly efficient immune cell reprogramming system. Cell Stem Cell 5, 554-566, doi:10.1016/j.stem.2009.10.004 (2009).

13) Di Tullio, A. et al. CCAAT/enhancer binding protein α (C/EBP(α))-induced transdifferentiation of pre-B cells into macrophages involves no overt retrodifferentiation. Proc Natl Acad Sci U S A 108, 17016-17021, doi:10.1073/pnas.1112169108 (2011).

14) Iritani, B. M. and Eisenman, R. N. c-Myc enhances protein synthesis and cell size during B lymphocyte development. Proc Natl Acad Sci U S A 96, 13180-13185 (1999).